



# Predicting probabilities of streamflow intermittency across a temperate mesoscale catchment

Nils H. Kaplan[1], Theresa Blume[2], Markus Weiler[1]

[1]Hydrology, Faculty of Environment and Natural Resources, University of Freiburg, 79098 Freiburg, Germany
[2]Hydrology, Helmholtz Centre Potsdam, GFZ German Research Centre for Geosciences, 14473 Potsdam, Germany

*Correspondence to*: Nils H. Kaplan (nils.kaplan@hydrology.uni-freiburg.de)

**Abstract.** The fields of eco-hydrological modelling and extreme flow prediction and management demand for detailed information of streamflow intermittency and its corresponding landscape controls. Innovative sensing technology for monitoring of streamflow intermittency in perennial rivers and intermittent reaches improve data availability, but reliable maps of streamflow intermittency are still rare. We used a large dataset of streamflow intermittency observations and a set of spatial predictors to create logistic regression models to predict the probability of streamflow intermittency for a full year, and, wet and dry periods for the entire 247 km² Attert catchment in Luxembourg. Similar climatic conditions across the catchment permit a direct comparison of the streamflow intermittency among different geological and pedological regions. We used spatial predictors describing land cover, track (road) density, terrain metrics, soil and geological properties as local as well as integral catchment information. The terrain metrics catchment area and profile curvature were the most important predictors for all models. However, the models which include the dry period of the year reveal the importance of soil hydraulic conductivity, bedrock permeability and in case of the annual model the presence of tracks (roads) during low flow conditions. A classification of spatially distributed streamflow intermittency probabilities into ephemeral, intermittent and perennial reaches allows the estimation of stream network extent under various conditions. This approach is a first step to provide detailed spatial information for hydrological modelling as well as management practice.

## 1. Introduction

Historically, streamflow observations and predictions have focused on perennial rivers. Even though intermittent streams and rivers may represent more than half of the global stream network (Datry et al., 2014) the study of these systems is much less abundant. Research on streamflow intermittency concentrated mainly on arid and semi-arid regions where these streams represent the dominant stream type due to the climatic conditions (Buttle et al. 2013). Intermittent streams in temperate regions have only recently gotten more attention (e.g. Buttle et al. 2013; Stubbington et al., 2017; Jensen et al., 2017, 2018, 2019; Kaplan et al., 2019; Prancevic and Kirchner, 2019). Streamflow intermittency in these regions may change in time dependent on seasonal climate conditions or in response to rainfall or snowmelt events (Buttle et al., 2013), whereas in the spatial



dimension it is controlled by the physiographic composition of the landscape, including geology, pedology, topography and land cover (Olson and Brouilette, 2006, Buttle et al., 2013, Goodrich et al., 2018, Jensen et al., 2018, Prancevic et al., 2019). Intermittency of streamflow, i.e. the drying and rewetting of streambeds, can be classified into ephemeral, intermittent and perennial by annual duration of streamflow (e.g. Hedman & Osterkamp, 1982; Matthews 1988; Jaeger & Olden, 2012), but

also based on hydrological processes including the spatial dimensions of hydrological connectivity (e.g. Sophocleous, 2002; Svec et al., 2005; Nadeau & Rains, 2007; Shanafield and Cook, 2014), or by ecological indicators (e.g. Hansen 2001; Leigh et al., 2015; Stromberg & Merritt, 2015). From a hydrological point of view the most consistent and frequently used classification of intermittency is based on the share of baseflow/groundwater contribution to total streamflow and is thus interrelated with the vertical and lateral connectivity between reach and groundwater (e.g. Sophocleus, 2002; Nadeau & Rains,

2007; Buttle et al., 2013; Godsey & Kirchner, 2014, Keesstra et al., 2018). Under regular conditions perennial streams gain groundwater throughout the year and maintain an almost permanent baseflow (Sophocleus, 2002). Thus, the groundwater table in perennial streams is above the level of the streambed throughout the year. In cold regions perennial streams can also be sustained from snowmelt (Nadeau & Rains, 2007). Intermittent rivers preserve continuous flow during certain times of the year when precipitation is high and/or evapotranspiration rates are lower and therefor the stream is receiving effluent

groundwater while in the dry season the streams loses water to the groundwater (Sophocleous, 2002; Zimmer et al., 2017). In ephemeral streams the groundwater table never reaches the level of the streambed, so influent groundwater conditions can only occur during flow events as direct response to strong rainfall or snowmelt events (Sophocleous, 2002; Zimmer et al., 2017). A stream can change the degree of intermittency along the channel and transition zones between geological parent materials can also cause abrupt changes in intermittency (Goodrich et al., 2018).

In contrast to the classification based on connection to the groundwater, the one based on streamflow duration is vague, because different climatic conditions result in climate specific proportional share of duration of streamflow presence throughout a year and thus lead to region-specific classification schemes (e.g. Hedman and Osterkamp, 1982; Hewlett, 1982; Matthews, 1988; Texas Forest Service, 2000). Hedeman and Osterkamp (1982) as well as Matthews (1988) classify streams as perennial when streamflow is present over 80 % of the time annually for the Western United States and the North American prairie respectively.

The threshold below which streams are classified as ephemeral ranges from 10 % to 30 % of the year so that the intermittent stream class has a range of bounding thresholds: more than 10-30% and less than 80%.
The spatial dynamics of streams and their longitudinal connectivity can be quantified by observing the streamflow continuity (temporal scale) and the longitudinal connectivity (spatial scale) with multiple sensors (e.g. EC-/temperature-sensors or time lapse imagery) along the stream (e.g. Goulsbra et al., 2009; Jaeger & Olden, 2012; Bhamjee et al., 2016, Kaplan et al., 2019),

or by mapping the wet stream network for several times at varying flow conditions (e.g. Godsey & Kirchner, 2014, Jensen et al., 2017). Despite of the existing classification schemes and advances in streamflow intermittency monitoring, accurate information of the spatial extent of intermittent stream network is sparse and often inaccurate (Hansen, 2001; Skoulikidis et al., 2017).





Recently this information gap is tackled with models to predict spatially distributed streamflow intermittency by using spatial predictors (Olson and Brouillette, 2006, Jensen et al., 2018, Prancevic et al., 2019) but also metrics that help to assess the longitudinal hydrological connectivity of rivers (Lane et al., 2009; Lexartza-Artza and Wainwright, 2009; Ali & Roy, 2010; Bracken et al., 2013; Habtezion et al., 2016). Prancevic et al. (2019) modelled the dynamical changes of stream network length

as a power function of the water discharge to the valley transmissivity. This transmissivity is represented through the topographic attributes slope, curvature and contributing drainage area. Olson and Brouillette (2006) used a logistic regression approach to differentiate between intermittent and perennial stream sites using a set of fifty basin characteristics as predictors. These included soil characteristics, geological grouping, mean elevation and land use as areal percentage of the contributing area but also terrain predictors like slope, relative relief and drainage area as well as climatological parameters like mean

annual precipitation. The logistic regression model approach from Jensen et al. (2018) focused on terrain metrics as predictors for predicting the probability of a stream being wet or dry. Most of the terrain metrics in their study were included as predictor on "local scale" as well as "mean upslope area". Among the most important predictors in these studies were topographic wetness index (TWI; Beven and Kirkby, 1979), topographic position index (TPI; Jensen et al., 2018), mean elevation, ratio of basin relief to basin perimeter, the areal percentage of well- and moderately well-drained soils in the basin (Olson and

Brouillette, 2006), drainage area (Olson and Brouillette, 2006, Prancevic et al., 2019), slope and curvature (Prancevic et al., 2019). The most successful predictors to model the spatiotemporal dynamics of the stream network are also part of the metrics developed to predict hydrological connectivity and are related to terrain (e.g. Lexartza-Artza and Wainwright, 2009; Ali and Roy, 2010), soil drainage and transmissivity (e.g. Nadeau and Rains, 2007; Lexartza-Artza and Wainwright, 2009; Ali and Roy; 2010). In addition, vegetation. land use and road network were investigated as control of hydrological connectivity (e.g.

Lexartza-Artza and Wainwright, 2009; Jencso and McGlynn, 2011; Bracken et al., 2013).

This study will build upon the work of Olson and Brouillette (2006) and Jensen et al. (2018) who aimed towards a separation of intermittent (dry) and perennial (wet) reaches using a logistic regression model (GLM) with a set of spatial predictors. In our study we present a new approach of using a GLM to predict not only the classes of intermittent/perennial or dry/wet but using probabilities of the model output to classify ephemeral, intermittent and perennial streams. Therefore, instead of using

binary data of e.g. intermittent (0) and perennial (1) the dependent variable in our models is the measure of relative intermittency, which represents the probability of streams having flow in a defined period (e.g. annual period) ranging between 0 and 1. In this way we can then classify the stream network into perennial, intermittent and ephemeral based on the statistical classification schemes (Hedman and Osterkamp, 1982). The set of predictors used in this study comprises land cover, road network, geology, pedology and terrain metrics both on "local scale" and "upslope area". The model was developed for the

mesoscale Attert catchment which catchment size of 247 km² ranges between those used in the studies of Olson and Brouillette (2006; 24.902 km²;) and Jensen et al. (2018; 0.7 to 0.12 km²).



## 2. Research area

The Attert River originates in the eastern part of Belgium and flows westwards into Luxembourg receiving its water from a catchment area of 247 km² at the outlet at Useldange (Hellebrand et al., 2008). The prevalent geologies of the catchment consist roughly from north to south of Devonian slate of the Luxembourg Ardennes (North West), sandy Keuper marls (centre) and

the Jurassic Luxembourg Sandstone formation (South) (Figure 1; Martínez-Carreras et al., 2012). Altitudes range from 245 m a.s.l. in Useldange to 549 m a.s.l. in the Luxembourg Ardennes. Lowlands with moderate relief dominate the topography in the Keuper marls with steeper slopes at the hilly Luxembourg Sandstone formation (Martínez-Carreras et al., 2012). Land use in the lowlands with Keuper marls is characterized by agriculture (41%) with considerable share of forest (29%) and grassland (26%) and small patches of urban areas (4%) while Sandstone areas are dominated by forest (55%) with lower proportions of

grassland and agriculture (39%). Land use in the slate dominated region in the Ardennes split into the plateaus which are predominantly used for agriculture (42%) and urban areas (4%) whereas the steep hillslopes and valleys are covered by forest (48%) and pasture (6%).

Soils in the Attert catchment include many of the major soil types of the temperate zone and are largely linked to lithology, land cover and land use (Cammeraat et al., 2018). Thus, dominant soils in regions with slate geology are stony silty soils

whereas the soils in the central parts of the catchment comprise mainly of silty clayey soils based on the Keuper marl geology and in the south sandy and silty soils are dominant in the Luxembourg sandstone formation (Müller et al., 2016). Cammeraat et al. (2018) pointed out the influence of land use on soil development in the Keuper marls with Stagnosols or Planosols under forest and Regosols under agriculture.

The climate in the Attert basin shows a strong impact of the westerly atmospheric circulation and temperate air masses from

the Atlantic Ocean which results in similar climate conditions for the whole basin (Pfister et al., 2017). Mean annual precipitation varies slightly from 1000 mm/a in the north-west to 800 mm/a in the south-east (Pfister et al., 2017) with a mean for the whole catchment of about 850 mm/a for the years 1971-2000 (Pfister et al., 2005). Seasonal changes in soil moisture and surface hydrology are induced by seasonal fluctuations of mean monthly temperatures (min. 0°C in January, max. 17°C in July) and thus, amount of monthly potential evapotranspiration (min. 13 mm in Dec., max. 80 mm in July) which

superimposes the low variability of monthly precipitation (min. 70 mm in Aug./Sep., max. 100 mm in Dec.-Feb; Pfister et al., 2005; Wrede et al., 2014).

Pfister et al. (2017) showed the strong impact of bedrock geology on the storage, mixing and release of water in the Attert catchment which determine the strong differences of seasonal flow regimes in areas of predominantly low-permeable bedrock (slate and marls) compared to permeable sandstone bedrock or diverse geology. Geology may also cause the strong differences

in the appearance of perennial and intermittent stream density which are visible in the topographic map (Le Gouvernment du Grand-Duché de Luxembourg, 2009). The catchment is subject to numerous anthropogenic alterations of surface flow. Surface and subsurface drainage, dams, ditches and river regulation measures changed the natural stream beds and flow conditions in the agricultural areas of the marly lowlands considerably. This can result in lower groundwater tables through drainage





measures and increased runoff velocity through e.g. straightened stream channels ultimately changing the periods with streamflow presence in ephemeral and intermittent streams (Schaich et al., 2011). Shifts in hydrological regime from intermittent to perennial can appear on the plateaus of the Ardennes, where some wastewater treatment plants are located (Le Gouvernment du Grand-Duché de Luxembourg, 2018).

**Figure 1: Geology and stream network of the Attert catchment and streamflow monitoring sites. Monitoring sites comprise "Sites" which were equipped with monitoring devices and "Virtual Sites" which were not permanently monitored but were never found to have surface runoff during several field trips in a 2 year period and thus are included as "zero-flow" virtual sites. Detailed maps show the more densely equipped areas in each predominant geology: slate (blue box), marls (red box) and sandstone (green box). The geological map from 1947 was provided by the Geological Service of Luxembourg (Adapted from Kaplan et al., 2019).**





## 3. Methods

### 3.1 Data

#### 3.1.1 Streamflow data

We used the data set of binary information of presence and absence of streamflow at 182 measurement sites in the Attert
5   catchment described and provided in Kaplan et al. (2019). The dataset combines streamflow data from various data sources
including time-lapse imagery, electrical conductivity sensors and water level measurements. Data from one year (July 2016 -
July 2017) with a temporal resolution of 30 minutes was used for this analysis. Sites were removed from the dataset if they
were (A) located downstream of the Attert gauge in Useldange, (B) contained extensive no data periods (> 50%) within the
selected one year period or (C) were located at positions where catchment calculations were not possible due to the relative
10  coarse resolution of the DEM. The data set analysed in this study comprises 164 sites of monitored intermittency.

The data set of the gauging sites is shown in Figure 2. In this study we model streamflow intermittency during a one-year
period on the one hand and two selected periods of 3 months (representing wet and dry conditions) on the other hand. The 164
sites chosen from Kaplan et al. (2019) contain 96 sites which show permanent flow and 50 sites with intermittent streamflow,
14 sites with ephemeral streamflow and one site indicating zero flow conditions throughout the year. The high share of sites
15  with perennial streamflow observations would lead to an overrepresentation of those sites in the statistical model. Thus, a total
of 21 virtual gauges with zero flow were added to the dataset in locations where numerous field observations over a two-year
period provide strong evidence of no surface flow conditions throughout the year. Hence the total number of sites used in this
study was 185 (Figure 1). The selection of the different modelling periods is based on the streamflow data and is closely related
to the often-used winter and summer seasons. Due to the extraordinary dry winter season the wet period is defined from
20  February to April whereas the dry period is defined from June to August, but consisting of the data from the years 2016 and
2017 due to the end of the available timeseries after July 2017 (Figure 2).

We introduce the measure of relative intermittency of streamflow $I_r$ as the ratio of the duration of streamflow occurrence to
the total duration of valid measurements in that given period:

$$I_r = \frac{\sum t_w}{\sum t_w + \sum t_d} \tag{1}$$

25  where $t_w$ are wet time periods with streamflow occurrence and $t_d$ are dry time periods without streamflow. Values between 0
and 1 represent the relative intermittency, with a value of 1 meaning continuous flow.



**Figure 2: Streamflow data used in this study. Gauge ID is a combination of the number on the left and the letters on the right. The data set combines data from different sources: Time-lapse camera (C), Conventional Gauges (CG) and Electric Conductivity measurements (EC). The wet and the combined dry period are indicated within the dark blue and orange boxes, respectivly. For the analysis of the dry period the summers 2016 and 2017 were combined. Discharge data (Q) at the outlet of the Attert catchment gauged in Useldange is shown at the top.**





### 3.1.2 Spatial data

#### 3.1.2.1 Contributing area averages

We tested a broad range of landscape feature data such as land use, topographical, pedological and geological properties with respect to their ability to predict $I_r$. Streamflow intermittency at a certain location is not only dependent on local characteristics

of the landscape represented by the pixel value of a raster layer at this location but also on the integral value of the upstream contributing area (CA). Therefore, the average value or proportion of landscape features in the contributing area was calculated for every cell of the associated landscape feature raster resulting in a new raster layer where every pixel value represents the average of the landscape feature of the contributing area. The SAGA GIS (version 2.3.2) tool "flow accumulation recursive" (Conrad et al., 2015) was used with the deterministic 8 algorithm and a DEM of 15 m resolution as elevation input to calculate

the number of contributing cells (output: catchment area) and the accumulated cell values of the landscape feature (output: total accumulated material) for all cells in the Attert catchment. We assume that all upstream cells contribute equally to the value of a pour point cell, thus, no weighting was included when accumulating cell values. The number of contributing cells $n$ was calculated from the contributing area divided by the cell size. Raster layers containing the landscape feature information (e.g. relative geological permeability, Figure 3) are used as input "material" $M_v$ for the "catchment area recursive" algorithm

and accumulate along the flow path through the catchment. The total accumulated material $M_t$ into a cell $xy$ for a given upslope area of $n$ cells can be written for each cell $xy$ as:

$$M_{t,xy} = \sum_{i=1}^{i=n_{xy}} M_v \tag{2}$$

The accumulated material $M_{t,xy}$ divided by the number of cells $n$ contributing to the cell $xy$ result in average values of the landscape feature in the sub-catchment:

$$\overline{M_{v,xy}} = \frac{M_{t,xy}}{n_{xy}} \tag{3}$$

Proportions of landscape features in a catchment result from a special case of catchment averages with $M_v$ values of 1 indicating the presence and values of 0 indicating the absence of selected landscape features.





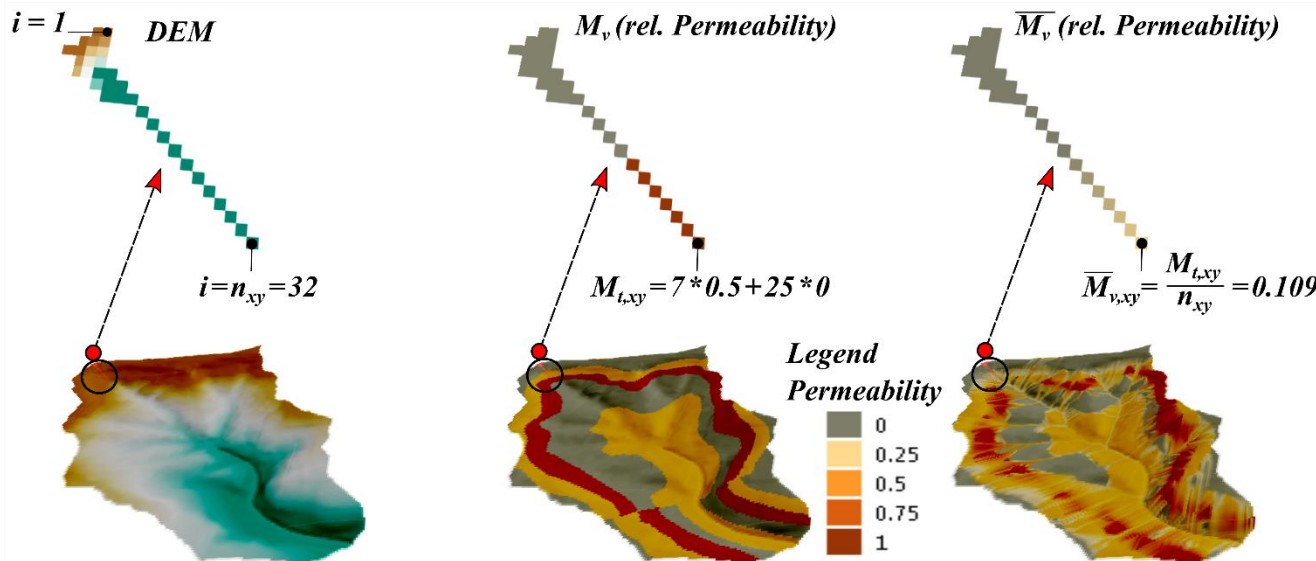

**Figure 3: Example for contributing area averages using relative bedrock permeability with values between 0 and 1. The Digital Elevation Model (DEM) and the relative bedrock permeability are used as inputs to calculate the catchment area averages. The example is calculated for the pixel *xy* at the uper left of the catchment (red line).**

### 3.1.2.2 Roads and tracks

The "highway" class from the open street map (OSM) dataset (OSM Wiki, 2018) was downloaded using the integrated OSM download function in QGIS. The dataset which includes all OSM-values featured under the OSM-key "highway" was split into a separate dataset containing only roads and a second one containing only tracks according to the OSM-values featured in

10  Table 1. The category "roads" represents the mainly sealed surfaces of the OSM-key "highway", whereas tracks characterize OSM-values of the "highway" key which usually are unsealed (Open Street Map Wiki, 2018).

**Table 1: Categories "streets" and "tracks" derived from OSM-key "highway".**

| Categories | OSM-Values in OSM-Key "highway" |
|---|---|
| Roads | Motorway, trunk, primary, secondary, tertiary (incl. link segments), unclassified, residential, service, living_street, pedestrian, raceway, bus_guideway, sidewalk, cycleway |
| Tracks | Track, escape, footway, bidleway, path |
| All Roads | All OSM-Values in "highway" |

15  Track densities around each cell [m/m²] of the different categories were calculated in ArcGIS for a radius of 25 m, 50 m and 100 m. The average track density per catchment was computed for all categories and radiuses using the approach of Eq. 3.





Manning's *n* for CORINE Landcover

Average Manning's roughness values were derived for all catchments based on the 2012 CORINE land cover dataset and land cover specific Manning's roughness values (Philips and Tadayon, 2006; Kalyanapu et al. 2009) by using Eq. 3. Table 2

5    provides an overview of the landcover classes and the respective Manning's values.

**Table 2: CORINE Landcover classes and their corresponding Manning's *n* values adapted from Kalyanapu et al. (2009)[1] and Philips and Tadayon (2006)[2].**

| CORINE Land Cover Class 1 | CORINE Land Cover Class 3 | Manning's *n* value |
|---|---|---|
| Forest | Broad-leaved forest | 0.036[1] |
| | Coniferous forest | 0.032[1] |
| | Mixed Forest | 0.04[1] |
| | Transitional woodland-shrub | 0.04[1] |
| Agriculture | Complex cultivation patterns | 0.031[2] |
| | Land principally occupied by agriculture, with significant areas of natral vegetation | 0.0368[1] |
| | Non-irrigated arable land | 0.030[2] |
| | Pastures | 0.0325[1] |
| Artificial surfaces | Discontinuous urban fabric | 0.00678[1] |
| | Mineral extraction sites | 0.00678[1] |

### 3.1.2.3 Terrain

Terrain analysis was based on a digital elevation model (DEM) with a grid size of 15 m and included catchment area ($A_c$), catchment height ($h_c$), catchment area volumes (CAV), slope, curvature, Topographic Wetness Index (TWI), Topographic Position Index (TPI), Vector Ruggedness Measure (VRM), Terrain Ruggedness Index (TRI) and the Mass Balance Index

15   (MBI).

Catchment area and height were computed with the SAGA GIS tool "catchment area recursive" (Conrad et al., 2015). Slope and curvature were computed using the corresponding tools from the ArcGIS 10.3 surface toolbox. Calculations for curvature comprise planar curvature (perpendicular to the direction of the maximum slope), profile curvature (parallel to the direction of maximum slope) and a combined measure of both planar and profile curvature (ESRI, 2020). Slope was also calculated as

20   catchment averages according to Eq. 3. Computations for Topographic Wetness Index (TWI) were based on the Topmodel approach which is accessible as SAGA-GIS Hydrology-toolbox with slope and catchment area as input. Topographic Position





Index (TPI, Guisan et al., 1999), Vector Ruggedness Measure (VRM, Conrad et al., 2015) and Terrain Ruggedness Index (TRI, Riley et al., 1999) were included as terrain roughness measures. All measures were determined with the SAGA-GIS Morphometry-toolbox and require the DEM data as input. Mass Balance Index (MBI, Friedrich; 1996, 1998) is a measure of landscape and sediment connectivity and was included as it can serve as a proxy for hydrological surface connectivity. MBI is

available from the SAGA-GIS Morphometry-toolbox.

Analogue to the hypsometric curve approach by Strahler (1952), catchment area volumes (CAV) represent the maximum possible upslope storage volume that can contribute to streamflow by gravimetric forcing. CAVs can either be calculated as a difference between surface and bedrock topography when focusing on soil processes or in a simpler approach as all material including bedrock and soil which is above and upslope of a given point in the catchment. We calculated the CAVs using the

second approach under the assumption that the main processes of transferring water through the volume to the outlet follows gravitational forcing and hence volume below the stream channel ($V_l$) does not contribute to water storage capacity through capillary or artesian processes. CAV was calculated in QGIS. In a first step, the average catchment elevation ($\bar{E}$) was calculated for all cells using Eq. 3. Second, subtracting the elevation which is equal or lower than the lowest position in the catchment (the pour point at cell xy, fig. 2) from its average elevation gives the average elevation of the catchment above the respective

outflow point which can be used to calculate the CAV as:

$$CAV = (\bar{E} - E_{min}) * A_c \tag{4}$$

with $E_{min}$ representing the minimal elevation of the catchment and $A_c$ as the catchment area.

### 3.1.2.4 Soil

Spatial information on soils is obtained from homogenized soil maps of Luxembourg and Belgium (see Table S1 in the Supplements). Homogenization was required due to slightly differing classification schemes in both national soil classification schemes. Available data includes information on soil texture, drainage behaviour and soil profile (see Table S2 in the Supplements). Saturated hydraulic conductivity ($K_s$) and field capacity ($\theta_a$) were derived from the homogenized soil maps and a set of soil hydrological parameters which is available from the combined field efforts of the CAOS research group

(Catchments as Organized Systems, see e.g. Zehe et al., 2014). Detailed information about the process is provided in the supplement section S1.

### 3.1.2.5 Geology

Spatial information of bedrock geology is based on a 1:25.000 scale geological map from 1947 provided by the Service

géologique de l'Etat (2018) in Luxembourg. Permeability classes were defined for all geological units and values of relative permeability assigned to each permeability class (Table 3). Relative permeability classes follow the approach of Pfister et al. (2017).



**Table 3: Classes of relative geological permeability for all geology units in the Attert catchment. Permeability classes were adapted from Pfister et al. (2017).**

| Geology | Permeability Class | Relative Permeability Value |
|---|---|---|
| Slates | Impermeable | 0 |
| Phyllades | Impermeable | 0 |
| Sandstone and Slates | Impermeable | 0 |
| Gypsiferous sandy marls | Impermeable | 0 |
| Gypsiferous marls (*groupe de l'anhydrite*) | Impermeable | 0 |
| Marls and sandstones (*Schistes de Virton*) | Impermeable | 0 |
| Marls and sandstones (*Formation de Mortinsart)* | Semi-permeable | 0.5 |
| Marls and dolomites (*Groupe de la Lettenkohle*) | Semi-permeable | 0.5 |
| Alluvial deposits | Semi-permeable | 0.5 |
| Silts with quaritic concretions (*Limons des Plateaux*) | Semi-permeable | 0.5 |
| Marls and Limestones (Formation de *Strassen*) | Semi-permeable | 0.5 |
| Marls and clay-limestones (*Elvange Formation*) | Semi-permeable | 0.5 |
| Shelly sandstone | Semi-permeable | 0.5 |
| Sandstones, clay and conglomerates | Semi-permeable | 0.5 |
| Dolomites and sandstones | Permeable | 1 |
| Luxembourg Sandstone | Permeable | 1 |

### 3.2 Statistical model

The relative intermittency data $I_r$ (Sec. 3.1.1) represents the likelihood of counts of the binary conditions flow or no flow, therefore this data can be modelled with a generalized linear model (GLM) using a quasibinomial link function. The quasibinomial link function is used to account for overdispersion. Spatial data described in section 3.1.2 was used as predictor dataset at all locations of the intermittency dataset (Table 4). Independence of predictors was checked by identifying linear correlation among the predictors. Predictors which showed no strong linear correlation with other predictors (threshold value

at 0.8, e.g. Famiglietti et al. 1998) were selected for the final model development and are listed in Table . Among highly correlated predictors first predictors were selected that were also highly correlated with multiple others and thus, reduce the predictor set by those highly correlated predictors. Secondly, if predictors of two main predictor classes were highly correlated, the predictor with the lower number of predictors in the class was chosen. The GLM model was derived from automated model selection using a stepwise backwards model selection approach based on the quasi Akaike Information Criterion (qAIC). GLM

and model selection were implemented in R software (R version 3.1.3) using the basic GLM functionality of R. In total 5 different models were developed: One model with intermittency data obtained from the entire time period of one year ("Model



Y", 01.07.2016 – 01.07.2017), two independent models whose predictor sets were selected based on the intermittency data from data subsets representing the wet ("Model W1", February-April) and dry ("Model D1", June-August) periods, i.e. with high and low flows observed in the streamflow data. Finally, two models based on the predictors selected by the "YEAR-Model" were set up and parameters and significance levels calculated by using the intermittency data of the wet ("Model W2")

and dry ("Model D2") periods instead of the annual period. Evaluation of the models Y/D2/W2 allows for direct comparison of parameter importance among all simulated periods and to test the applicability of the predictor selection from the Model-Y to the wet and dry periods of the modelled year.

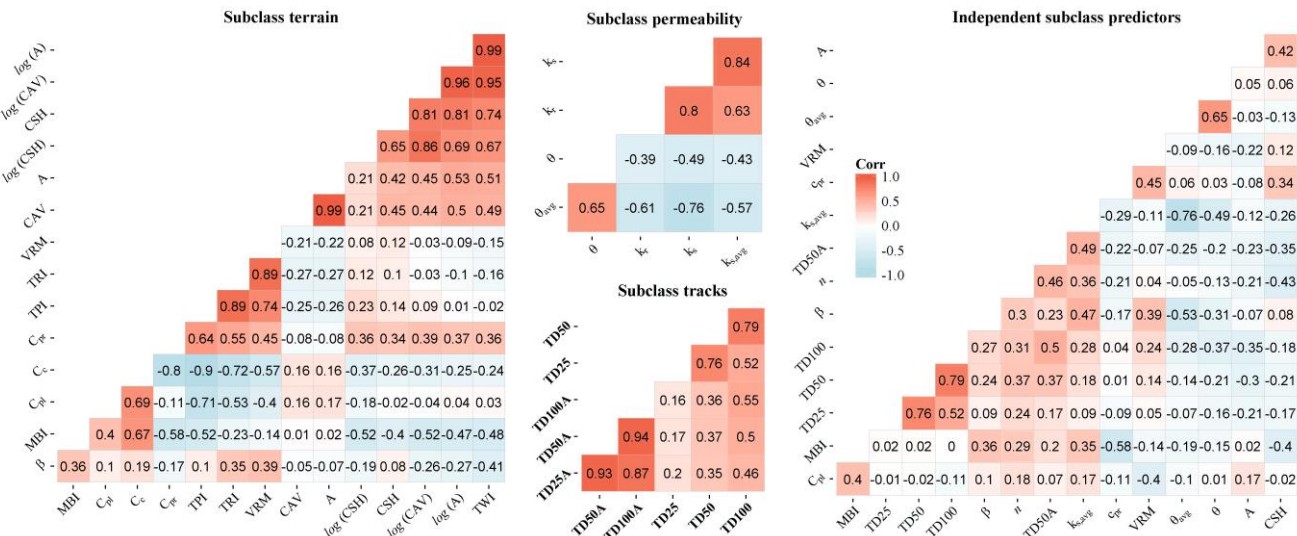

**Figure 4: Correlations between predictors. Correlations are first shown for each subclass (left, mid) and the correlation of the independent predictors of all subclasses after final selection. Predictors which show a correlation coefficient ≤ 0.8 were selected from the subclasses. From strongly correlated predictors (correlation coefficient ≥ 0.8) those were selected which can be derived from basic analysis of the geospatial data. Characteristics that combine multiple predictors such as TWI (combination of slope and catchment area) were preferably rejected as predictors when strongly correlated to their corresponding combinations. The final**
**predictor set of independent (not strongly correlated) predictors is shown on the right side.**



**Table 4: Predictors and their abbreviations for GLM development. All predictors are based on the available geodata. The scale of the predictors indicates whether the predictors were calculated on local scale (at the pixel scale) or represent an integral measure of the contributing area according to Eq. 3. Predictors with correlation coefficients of ≤0.8 (Figure 4) and a selection of the most representative predictors among the highly correlated predictors were included in the final model development and are written in bold.**

| Predictor Main Class | Predictor Sub-Class | Abbreviation | Scale |
|---|---|---|---|
| *Road Network* | **Track Density 25 m Radius** | **TD25** | **Local** |
| | **Track Density 50 m Radius** | **TD50** | **Local** |
| | **Track Density 100 m Radius** | **TD100** | **Local** |
| | Track Density 25 m Radius average of contributing area | TD25A | contributing area |
| | **Track Density 50 m Radius average of contributing area** | **TD50A** | **contributing area** |
| | Track Density 100 m Radius average of contributing area | TD100A | contributing area |
| *Landuse* | **Manning's $n$** | **$n$** | **contributing area** |
| *Soil* | **Effective Saturated Hydraulic Conductivity** | **$K_{s,avg}$** | **contributing area** |
| | **Field Capacity** | **$\theta$** | **contributing area** |
| | **Catchment Average Field Capacity** | **$\theta_{avg}$** | **contributing area** |
| *Geology* | **Relative Bedrock Permeability** | **$K_r$** | **contributing area** |
| *Terrain* | ***log*(Catchment Area)** | **A** | **contributing area** |
| | Catchment Area Volumes | CAV | contributing area |
| | **Catchment Storage Height** | **CSH** | **contributing area** |
| | **Catchment Average Slope** | **$\beta$** | **contributing area** |
| | **Curvature Planar** | **$C_{pl}$** | **Local** |
| | **Curvature Profile** | **$C_{pr}$** | **Local** |
| | Curvature Planar & Profile Combined | $C_c$ | **Local** |
| | Topographic Wetness Index (Topmodel) | TWI | Local |
| | Topographic Position Index | TPI | Local |
| | **Vector Ruggedness Measure** | **VRM** | **Local** |
| | Terrain Ruggedness Index | TRI | Local |
| | **Mass Balance Index** | **MBI** | **Local** |





The importance of predictors was determined by the automated selection based on the qAIC. The significance of each predictor for the model is rated through the p-values of the GLM-output. The model performance was analysed based on the Nagelkerke peudo-$R^2$ measure in order to evaluate an overall model fit but also for the ability of each model to predict intermittency classes ranging from ephemeral over intermittent to perennial. McFadden pseudo-$R^2$ was used as performance measure for all GLMs.

The observed and modelled data was classified according to the degree of intermittency into ephemeral ($I_r < 0.1$), intermittent ($0.1 \leq I_r < 0.8$) and perennial ($I_r \geq 0.8$). We used the same classification classes to describe the degree of intermittency for comparison of the three-month periods used to model the wet and dry period although the terms ephemeral, intermittent and perennial apply solely for the annual period.

## 10    4. Results

### 4.1 Predictor importance

The results of all models show that the most important predictors for modelling relative intermittency are the logarithm of the catchment area *log(A)* and profile curvature $C_{pr}$ (Table 5). The predictors of soil saturated hydraulic conductivity, bedrock permeability and track density become important when modelling the dry period. Apart from are the logarithm of the catchment

area *log(A)* and profile curvature $C_{pr}$ as most significant predictors, the predictor set for Model-Y also included effective saturated hydraulic conductivity and relative bedrock permeability, but with lower significance levels (Table 5). Track density within a 100m radius was only selected for Model-Y and contributes to the model on a rather low level of significance. The predictors found in the Model-Y were used for the models W2 and D2 and showed differing significance for these two periods. While log(A) and $C_{pr}$ had a significant contribution to both models, the predictor of saturated hydraulic conductivity and

bedrock permeability were only significant for the dry period (Table 5). Track density was not important in either of the two sub-periods. On the other hand, based on the full set of available predictors the automated model selection process for the models W1 and D1 only chose those predictors which were also significant in the corresponding models W2 and D2 without adding additional predictors from the overall set of predictors. For the wet period a predictor set including profile curvature and catchment area on log-scale was identified, while for the dry period saturated hydraulic conductivity and bedrock

permeability were added to the predictor set, resulting in a small increase of explained variance (Table 5).





**Table 5: The significance of each predictor for each model. The intermittency values of the Model-Y was based on an annual period of flow observations whereas the models W1 and W2 represent the three wettest month of the annual period and the models D1 and D2 the three driest months. Significance codes represent the following P-Values for model predictors: 0.000 = *** ; 0.001 = ** ; 0.01 = * ; 0.05 = L; not significant = x. Positive and negative signs indicate the signs for the model parameter estimations.**

| Parameter | Model-Y | D1 | W1 | D2 | W2 |
|-----------|---------|-----|-----|-----|-----|
| Intercept | - (***) | - (***) | - (***) | - (***) | - (***) |
| $C_{pr}$ | + (**) | + (*) | + (***) | + (*) | + (**) |
| A | + (***) | + (***) | + (***) | + (***) | + (***) |
| $K_{s,avg}$ | + (*) | + (*) | | + (*) | + (x) |
| $K_r$. | - (*) | - (L) | | - (*) | - (x) |
| TD100 | - (L) | | | - (x) | - (x) |

**Table 6: Explained individual variance (McFadden pseudo-R²) of the models with predictors added to the model starting from a single predictor model using *ln*(Catchment Area) with the lowest pseudo-R².**

| Parameter | Model-Y | W1 and W2 | D1 and D2 |
|-----------|---------|-----------|-----------|
| A | 0.148 | 0.130 | 0.111 |
| $C_{pr}$ | 0.159 | 0.147 | 0.120 |
| $K_{s,avg}$. | 0.160 | 0.148 | 0.121 |
| $K_r$. | 0.164 | 0.151 | 0.126 |
| TD100 | 0.168 | 0.153 | 0.127 |

### 4.2 Model performances

Considering the McFadden pseudo R² between 0.2 and 0.4 for a good model fit (Backhaus et al., 2006), low values for pseudo-R² were found for all GLMs, ranging between 0.147 (W1) to 0.168 (Model-Y, Table 6) Nonetheless, the error matrix based on the classified data reveals the ability of the model to correctly classify the intermittency classes of ephemeral, intermittent and perennial sites (Table ). The Model-Y shows 59% correct classifications for intermittent streams and 89% for perennial streams. Ephemeral streams are not well represented by the model with only 18% correct classifications (Table 7). For the

Models W1 and W2 80% of the intermittent, 21% (23%) of the ephemeral and 86% (83%) of the perennial stream sites were correctly classified. Similar performances show both Models D1 and D2 with 33% (29%) correct classifications for ephemeral, 67% for intermittent and 70% for perennial stream sites.

The overall accuracy for the modelled intermittency classes is increasing with a higher number of monitoring sites having permanent streamflow and is within the range of 58-60% for the dry period, to 68% for the year period and 72-73% for the

20 wet period. Correct classifications depend strongly on relative bedrock permeability with low classification performance for





sites with high bedrock permeability and higher performance for sites with low bedrock permeability (Figure 5). The number of monitoring sites with ephemeral streamflow is low compared to the sites with intermittent and perennial streamflow (Figure 6). In contrast to the observations the number of modelled ephemeral streams is overestimated by all models as the modelled intermittency values show a strong tendency towards the extreme values of flow or zero-flow (Figure 6).

**Table 7: Confusion matrix for the stream classification of ephemeral, intermittent and perennial. Counts within each class are shown in bold, the percentage of modelled class counts within each measured class are shown in brackets. Grey shaded values highlight the correct predictions for each class.**

| | | Measured Intermittency | | |
|---|---|---|---|---|
| | | Ephemeral | Intermittent | Perennial |
| | | Model-Y | | |
| Simulated | Ephemeral | **7** (**18%**) | **1** (3%) | **0** (0%) |
| | Intermittent | **31** (79%) | **22** (**59%**) | **12** (11%) |
| | Perennial | **1** (3%) | **14** (38%) | **97** (89%) |
| | | Model W1 | | |
| | Ephemeral | **7** (21%) | **0** (0%) | **1** (1%) |
| | Intermittent | **25** (73%) | **24** (80%) | **16** (13%) |
| | Perennial | **2** (6%) | **6** (20%) | **104** (86%) |
| | | Model W2 | | |
| | Ephemeral | **9** (24%) | **0** (0%) | **1** (1%) |
| | Intermittent | **22** (65%) | **24** (80%) | **19** (16%) |
| | Perennial | **3** (9%) | **6** (20%) | **101** (83%) |
| | | Model D1 | | |
| | Ephemeral | **16** (33%) | **0** (0%) | **0** (0%) |
| | Intermittent | **32** (67%) | **18** (67%) | **33** (30%) |
| | Perennial | **0** (0%) | **9** (33%) | **77** (70%) |
| | | Model D2 | | |
| | Ephemeral | **14** (29%) | **0** (0%) | **1** (1%) |
| | Intermittent | **33** (69%) | **18** (67%) | **32** (29%) |
| | Perennial | **1** (2%) | **9** (33%) | **77** (70%) |



Figure 5: Measured intermittency is plotted against modelled intermittency for each model. Relative bedrock permeability is color coded. The grey boxes indicate the classes of ephemeral, intermittent and perennial streamflow.

**Figure 6: Distribution of modelled and measured intermittency for each model. The measured intermittency values show a strong trend towards the higher intermittency values and contain for the year and for the wet models very low numbers in the zero-flow intermittency bin. The modelled intermittency values show a strong tendency towards the minimal and maximum intermittency values.**

## 4.3 Prediction Maps

Intermittent and perennial streams were predicted for the entire Attert catchment based on spatially distributed predictor data (Figure 7). All modelled stream networks have a tendency to show many more first order streams compared to the stream network of the topographic map (Le Gouverment du Grand-Duché de Luxembourg, 2009). The model also predicts streams in areas of agricultural land use where the topographic map shows no streams. The W1 model set up for the wet period is driven





by two predictors catchment area and curvature. The additional predictors in the W2 lead to a large increase of the modelled stream length of the intermittent streams (Table 8) which becomes visible in the mapped stream network with a high density of intermittent streams in areas of lower geological permeability (Figure 7). However, models for the dry period generally show lower numbers of first order streams compared to the other models (Figure 7) and thus, also the length of the intermittent

stream network is also in higher agreement with the topographic map (Mapped Streams, Table 8). Though, the models for the dry periods underestimate the length of the perennial stream network compared to the topographic map (Table 8). Expansion of the stream network with the change from dry to wet period becomes visible through the stream length of the modelled stream networks (Table 8). The total stream length for the dry period models are 684 and 833 km while the stream length for the models W1 and W2 ranges from 1317 to 2109 km. On average the modelled perennial stream network expands with a

factor of 1.4 from dry to wet period while the intermittent streams show a change in stream length of a factor 2.5. Stream length of the perennial streams in Model-Y is with 227 km within the range of the mapped perennial stream length with 274 km. However, the intermittent stream length is with 658 km for the Model-Y eight times higher than the mapped stream length with 82 km (Table 8).

**Table 8: Stream length [km] of modelled streams and mapped streams from the topographic map (Le Gouvernment du Grand-Duché de Luxembourg, 2009).**

| | Modelled Stream Length [km] | | |
|---|---|---|---|
| Model | Perennial Streams | Intermittent Streams | Total |
| Model-Y | 227 | 658 | 885 |
| W1 | 246 | 1071 | 1317 |
| W2 | 278 | 1831 | 2109 |
| D1 | 179 | 505 | 684 |
| D2 | 191 | 642 | 833 |
| Mapped Streams | 274 | 82 | 356 |





**Figure 7: Prediction maps of intermittency. Ephemeral streams are not displayed. Intermittent streams are defined for streamflow present between 10% and 80% of the modelled time period, perennial for streams with ≥80% streamflow presence.**



## 5. Discussion

### 5.1 Evaluation of GLM model predictors

Intermittency of rivers results from superimposed interactions among climatic factors (ET, P), the physiographic layout of the landscape (geology, topography, topology, soil type, land cover) and possible artificial alterations (streets, land use, drainage,

water supply) (Buttle et al., 2013; Costigan et al. 2016; Jaeger et al., 2019). Some of the physiographic attributes can be expressed in physically meaningful, yet simplifying representation, e.g. spatial information of hydraulic conductivity in soils simplifies soil heterogeneity and presence of macropore flow (van Genuchten, 1980; Weiler & McDonnell, 2007). For other predictors classified representations are necessary due to difficulties to gather representative data on larger scale. This applies to the hydraulic conductivity in bedrock represented in this study as relative geological permeability (Pfister et al., 2017) or

terrain metrics such as terrain roughness which provide a measure for sources and sinks at the surface (Ali & Roy, 2010; Bracken et al., 2013; Boulton et al., 2017).

We assume for the selection of predictor variables in this study that climatic heterogeneity plays a minor role in our catchment, which is supported by the small differences in annual precipitation (Pfister et al., 2005; Wrede et al., 2014). Focusing on non-climatic predictors we find a general importance of the contributing area and profile curvature among all models tested. This

finding is consistent with the studies of Prancevic & Kirchner (2019) who predicted the extension and retraction of stream networks based on the topographic attributes slope, curvature and contributing drainage area.

The topographic wetness index (TWI) is frequently used as topographic attribute to predict streamflow permanence at the local scale and the extent of the perennial stream network (Hallema et al., 2016; Jensen et al., 2018; Jaeger et al., 2019). However, the TWI was not included as important predictor due to its high correlation (r = 0.99) with contributing area on log-scale. Thus,

in this study the TWI is represented through the combination of catchment area and curvature, which was confirmed by a test run for model selection using the TWI instead of contributing area.

Other important predictors include the hydraulic conductivity of the soil and the relative bedrock permeability both as integral measure for the contributing area. The importance of bedrock permeability was emphasized by Pfister et al. (2017) who identified bedrock permeability as major control for storage, mixing and release of water in the Attert and Alzette River basin.

Both predictors control the storage of water in the catchment (Buttle et al., 2013; Pfister et al., 2017) and the transit time (Costigan et al., 2016; Zimmer & McGlynn 2017; Pfister et al., 2017) of water through the catchment. Generally, storage capacity of water in the catchment can determine the permanence of water availability and thus the permanence of flow. Also, the potential velocity of surface and subsurface flow facilitated by the catchment properties can have a direct impact on flow permanence.

According to Prancevic & Kirchner (2019) data of width, thickness and conductivity for the permeable zone underlying temporal channels is not available. They therefore derive the parameter valley transmissivity which represents a combination of bedrock, soil permeability and the valley cross-sectional area from topographic attributes. Thus, the most important predictors identified in our study are in strong agreement with those used by Prancevic & Kirchner (2019) who model the





extension and retraction of flowing streams and the study of Naedeau & Rains (2007) on initiation of fluvial erosion. Besides the transmissivity of the soil and bedrock, the infiltration capacity of the surface can cause surface flow initiation. Not only paved surfaces but also logging tracks were identified as source areas of Hortonian overland flow (Ziegler & Giambelluca, 1997).

In our study, the density of tracks in a 100m radius was identified as a predictor in the model for the annual period showing the potential importance of the low infiltration capacity of tracks during strong precipitation events. However, this predictor had no importance for the other periods. This could be attributed to the low proportion of tracks in the catchment with sufficient inclination to cause Hortonian overland flow. Additionally, most of the observed logging tracks are located in a geological setting with sandstone bedrock and sandy soils. Thus, observed events in the dry periods are limited to a low number of storm

events with sufficient precipitation to generate surface runoff. Due to the very short time with flow these sites may reduce their weight in the automated predictor selection compared to no-flow sites. Nonetheless, for individual tracks Hortonian overland flow initiation can be important (Ziegler & Giambelluca, 1997).

The use of integral information of averaged predictor values based on contributing area was helpful to predict point scale intermittency, although, abrupt changes of intermittency due to local scale geological layout have been reported by e.g.

Goodrich et al. (2018). Geological permeability and hydraulic conductivity were included as averaged information of the catchment, while curvature and track density are point scale information. Although integral and point scale information are strongly correlated at the sites of this dataset the model benefits not only from the lower correlation among the predictors with integral information of geological permeability and soil hydraulic properties. By using integral predictors, we take into account that the streamflow intermittency at any point in the catchment can be influenced by the overall contributing area properties

(see e.g. Olson and Brouilette 2006, Pfister et al., 2017, Jensen et al., 2018). Streamflow initiated upstream will be maintained when the longitudinal hydrological connectivity allows the propagation of the flow downstream. Therefore, vertical or lateral connectivity measures which are also strongly linked to permeability (Jensco et al., 2010; Boulton et al., 2017) need to be considered as integral component of the catchments that contributes to the probability of streamflow. The integral information of geological permeability and soil hydraulic conductivity may be able to serve as one of these measures.

## 5.2 Variability and uncertainty in model predictions

Spatially distributed model predictions of streamflow probabilities enable the comparison of model output with the mapped stream network from the topographic map covering the diverse geologies, soils, land cover and topography in the Attert catchment. Classification based on streamflow intermittency separates stream reaches into ephemeral, intermittent and

perennial streamflow classes to derive a hierarchical stream network containing the intermittent and perennial reaches (Figure 6). We are aware of the fact that the number of gauging sites limits the model evaluation with a split calibration-validation approach. We used 172 sites to develop the GLMs with up to five predictors which is within the range of necessary 20 to 50 observations per variable proposed by van der Ploeg et al. (2014) for a good GLM setup. As the distribution of measurement





sites in the data set has a strong tendency towards permanent streamflow sites and thus to the perennial reaches, this leads to an underrepresentation of the intermittent and ephemeral reaches in the data when splitting the data for model evaluation (Figure 6). We therefore evaluated the model by its ability to predict the spatial distribution of intermittent/perennial streams compared to the mapped stream network. We assume that the mapped stream network approximately represents the natural

layout of the stream network in areas with lower human impacts. However, alteration of the natural stream network in areas of artificial and agricultural land use can be severe and thus misleading when comparing to model results.

Changes between wet and dry periods of the year result in expansion and contraction of the stream network (Buttle et al., 2012). This process is predicted in the model results of the changes in stream length of perennial and intermittent streams (Table 8). We use the classification of perennial and intermittent streamflow for all modelled periods to use a consistent

classification although we are aware that the original definition is based on annual streamflow and does not address the streamflow intermittency of a 3-month period. Perennial here simply means that streamflow is permanent over the 3-month period. The accuracy of class predictions of perennial and intermittent streams varies significantly between the time periods used for the model setup (Table 7). Predictions of intermittent and perennial streams during the dry season are fairly well represented by the model. This goes hand in hand with a reduced number of predictors in the model with solely the two

topographic predictors profile curvature and contributing area. The dominant role of terrain metrics, which are highly correlated with the TWI, reflects the importance of runoff generation processes leading to saturation and maintaining streamflow in wet conditions. Those processes include the rise of the groundwater table and high soil saturation during the wet season which enhance the vertical and lateral hydrological connectivity (Hallema et al., 2016; Zimmer er al, 2017; Keesstra et al., 2018).

The comparison between models for the wet, dry and annual period reveals the additional complexity in the system as additional predictors are necessary to predict the wet system state. Model accuracy for classes with intermittent and perennial streamflow decreases slightly for models of the dry and annual period in comparison to the models for the wet period. Conversely, model accuracy for the ephemeral class increases. However, for the wet period, model accuracy of the intermittent and ephemeral classes is directly linked to the low number of sites that cease to flow during the wet period. The shift of the

observed data towards conditions of perennial flow and the underrepresentation of intermittent sites leads to lower model accuracies for the models W1/W2.

All models have a general tendency to overestimate the extremes of relative intermittency classes close to zero and permanent flow (Figure 5). Simulated intermittent stream length increases by 112 to 185 % between dry and wet model periods, whereas perennial stream length increases by 37 to 45 %. Prancevic et al. (2019) calculate a hypothetical change in stream length

between 10 % for a low and 900 % for a highly dynamic stream network using similar model predictors as in this study. To increase the low predictive power of the ephemeral and intermittent model classes additional sites with information of sustained no-flow conditions could enhance the predictive power for these classes.

Bedrock permeability of the catchments is a major control of the hydrology of the catchment and is also identified as major predictor for the annual and dry period models. Nevertheless, catchments with high geological permeability lack proper





representation by the model particularly for sites with low streamflow intermittency (Figure 4). One data-inherent reason for the low model accuracy in catchments with highly permeable geology results from the lower number of sites representing such geological condition. Process based reasons arise from the geological setup which is needed for the initiation of sources in the highly permeable geologies of Buntsandstein and Luxembourg Sandstone in the Attert. Springs were observed to be initiated

at the boundary of rather impermeable marls and the thick layer of overlaying highly permeable sandstone. They usually maintain the perennial reaches in these catchments throughout the year due to large dynamic storages (Pfister et al., 2018). Thus, for predictions not only the information of the mean geological permeability of the bedrock is needed but also the thickness and orientation of subsurface layers differing in permeability. Less permeable geologies are better represented in all models (Figure 4) but would also benefit from a larger number of sites of intermittent streams to enhance the model accuracy

for this class. Intermittent streams turned out to be more important in areas with less permeable geologies. This could result from smaller storage capacity which is not able to maintain perennial streamflow throughout the year in the marl and slate geologies of the catchment (Pfister et al., 2018). Intermittency in the marl geology can also be induced by land use. The modelled stream length of intermittent streams is significantly higher than the mapped streams of the topographic map (Table 10). The maps in Figure 6 reveal key areas with agricultural land use that contain substantially more modelled intermittent

streams than the topographic map. The modelled streams may not be completely wrong when assuming a natural environment but streamflow in these areas was heavily altered by artificial surface and subsurface drainage (Schaich et al., 2011). Sites which are located in catchments with merely agricultural land use are underrepresented in our dataset. Thus, a higher spatial density of these sites may improve the representation of such areas.

The predictors for hydraulic conductivity were derived from multiple soil maps and translated the soil attributes to saturated

hydraulic conductivity and field capacity. Although deriving hydraulic properties from texture information using pedo-transfer functions is a common procedure (Wösten et al., 2001), spatial information of transmissivity in valleys based on hydraulic conductivity of soil and bedrock is often not available for all soils and rock formations in the area of interest (Prancevic et al., 2019). We tried to capture the effects of soil heterogeneity on permeability as much as possible by including factors that alter effective hydraulic conductivity such as soil drainage (Clausen and Pearson, 1995) and soil horizons (Zimmer and McGlynn,

2017). This required some assumptions for the parametrization of the soil maps which needed to be based on sparse data from literature and a small database of soil properties from the research area. These assumptions potentially introduce uncertainty to the effective saturated hydraulic conductivity. Nonetheless, this data adds valuable information to the soil hydraulic properties and their representation in the statistical models. The predictor of relative geological permeability relies strongly on the classification of the underlying dataset. The dataset provides only a coarse classification of geological permeability and

misses information of geological layering. Nonetheless, the permeability data both for soil and bedrock are crucial information to predict streamflow intermittency with in our models.

Further uncertainty in the predictions may arise from the quality of the geospatial predictor data. Terrain metrics are dependent on the quality and the resolution of the underlying DEM (Habtezion et al., 2016). In this study only a DEM with 15 m spatial resolution was availale to derive terrain metrics (e.g. contributing area, slope, curvature, TWI) which allowed delineation of





most streams. However, some small channels in flat areas such as road ditches or tile drainages require a higher resolution of the DEM to calculate the exact terrain metrics in such areas. Coarser DEMs enhance hydrologic connectivity by reducing depression storage and therefore increase the probability of runoff (Habtezion et al. 2016). Thus, terrain predictors require DEMs with particular small cell-size when aiming for an adequate representation of intermittent and ephemeral reaches in

models. Using a coarse cell-size DEM can result in a shift of sites into larger catchments which are actually located in smaller catchments in cases where accuracy of the site's position is lower than the cell-size of the DEM. With a maximum spatial deviation of 8 m for the site position, mismatching between sites and cells can occur. With contributing area and curvature two predictors of the GLMs are dependent on DEM resolution and are prone to the discussed errors. Contributing area can be either overestimated or underestimated caused by inaccurate localisation of sites and the coarse cell size of the DEM.

Misrepresentation of curvature can be caused from coarse cells that submerge micro-topographic information. Therefore, a DEM with smaller cell-size (2-3 m) can enhance model results and better representation of reaches with low relative intermittency (Habtezion et al. 2016; Jensen et al., 2018). However, such a finer resolution DEM was not available for the study area.

The simulated performance of the GLMs is generally low compared to other studies which use GLMs to discriminate between

intermittent and perennial streamflow (e.g. Olson and Brouilette, 2006; Jensen et al., 2018). The low performance arises from the higher model complexity with the aim to model relative intermittency instead of discriminating only between the two classes of intermittent and perennial streams.  In addition, the dataset used in this study is limited to point measurements instead of mapped stream reaches. Missing the complete information along the stream complicates also to trace the movement of channel heads over time. Thus, the highly dynamic transitions of streamflow intermittency at the most upstream sections of a

reach are neither represented by the data nor can it reflect the sharp transition zone to areas with no-flow. The missing information of exact position of the channel heads is also leading to an overestimation of the length of the intermittent stream network (Fig. 6). This can be improved by defining areas of zero-flow when observing flow occurrence throughout the seasons (with e.g. time-lapse camera) and especially during strong precipitation events (e.g. visual observations). However, the model results for the three intermittency classes are promising and the performance of the model could benefit from denser monitoring

networks and extended field observations mainly of sites with intermittent to no flow. Thus, our modelling approach advances from previous studies that used GLMs to discriminate between perennial and intermittent streamflow by adding the ability to discriminate between the full range of probabilities between zero- and permanent flow (e.g. Olson and Brouillette, 2006; Jensen et al., 2018).

## 6. Conclusion

This study presents a novel approach of modelling streamflow intermittency using logistic regression models with data of streamflow presence/absence and spatial predictors. Significance and selection of model predictors varied among models of wet and dry periods, indicating a change of predictor importance for wet and dry states of the catchment. Models for the wet





periods were mainly driven by the terrain metrics contributing area and profile curvature. The high correlation of catchment area on log scale to the topographic wetness index (TWI) indicates that the probability of saturation is an important driver in wet periods. Additional predictors of saturated conductivity and relative bedrock permeability that define the transmissivity and the storage capacity of the system as well as the track density as potential indicator of local Hortonian overland flow reveal

a system of higher hydrological complexity during periods of low streamflow. Furthermore, the selection of predictors shows the viability of the innovative approach using integrated contributing area information for certain predictors such as effective saturated conductivity and relative bedrock permeability. Both predictors contribute significantly to the models for the dry periods. This indicates that the loss of water through high infiltration capacity and storage capacity in the upstream contributing area are among the controlling factors of streamflow intermittency.

Modelling results classified into ephemeral, intermittent and perennial streamflow are promising, yet the overall modelling accuracy needs to be improved by denser spatial information of streamflow intermittency ground truth and digital terrain models of higher resolution. Modelling results are classified into ephemeral for streamflow presence of less than 10%, intermittent between 10% and 80% and perennial for more than 80% within a defined time period. Based on this classification all models are able to discriminate between intermittent and perennial streams. Changes in length of the stream network when

shifting from wet to dry state of the catchment are captured by the models but correct representation of the whole stream network was not yet achieved. Future testing the model in catchments of different sizes and climates with a higher data density could improve the classification thresholds and cumulate in a comprehensive and representative classification. A logistic regression model approach as presented in this study has the potential to provide the information of the streamflow probabilities throughout the year, but also for the wet and dry state of a catchment and therefore the dynamics of the stream network rather

than a static stream network. The logistic regression model is simple to set up and can be trained with different predictor sets. We recommend a larger sample size for model application to achieve reliable modelling results. Maps of streamflow probability are rare but would be extremely beneficial for ecological modelling, operational implementation of water policies for catchment conservation and regulation as well as modelling of flash flood induced streamflow. The share of streams with non-permanent streamflow within the total stream network and the spatial extent is critical information for researchers as well

as for river ecosystem- and extreme-event management.

**Author contributions**

Nils Kaplan prepared the data, designed the analysis and carried it out. Nils Kaplan prepared the manuscript with contributions from all co-authors.



**Competing interests**

The authors declare that they have no conflict of interest.

**Acknowledgements:**

This study was funded by the German Research Foundation (DFG) within the Research Unit FOR 1598 Catchments As Organized Systems (CAOS) – subproject G "Hydrological connectivity and its controls on hillslope and catchment scale stream flow generation". The article processing charge was funded by the Baden-Württemberg Ministry of Science, Research and Art and the University of Freiburg in the funding programme Open Access Publishing. We thank the Luxembourg Institute of Science and Technology (LIST) for providing geospatial and discharge data. We appreciate the work of Cyrille Tailliez and

Jean François Iffly who are responsible for the installation, maintenance and data processing at the LIST and contributed with their work to our project. We  thank Ernestine Sohrt, Uwe Ehret and Conrad Jackisch for providing the initial homogenized soil map as well as Dominic Demand, Jérôme Juilleret and Christophe Hissler for helpful information about the soils in the Attert catchment. The underlying streamflow intermittency data is available at http://doi.org/10.5880/FIDGEO.2019.010 and is described in detail by Kaplan et al., (2019).

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
