# Peer review of "Predicting probabilities of streamflow intermittency across a temperate mesoscale catchment"

_Hydrology and Earth System Sciences, 2020_

## Referee Comment (RC1) · Anonymous Referee #1 · 24 Jun 2020

General comments:

In "Predicting probabilities of streamflow intermittency across a temperate mesoscale catchment," Kaplan et al. explain the local and accumulated catchment controls on flow intermittence along the flow network of the Attert catchment in Luxembourg. Using logistic regression models for annual as well as wet and dry periods, the authors evaluate the variable importance of land cover, road density, soil, geology, and terrain metrics in controlling flow intermittence. The authors use a unique, empirical high spatial and temporal dataset (Kaplan et al., 2019) to develop these models. The authors presentation of model results and discussion of implications and uncertainties is fairly robust but with several opportunities for enhancement prior to publication. In addition, there are grammatical and tense errors and inconsistency throughout, so the

manuscript needs detailed proofreading prior to publication. This study is an important scientific contribution with high scientific and presentation quality.

Specific comments:

- Introduction does a good job of describing prior work and science gaps in the field of hydrology relevant to studying drivers of intermittent flow occurrence as well as spatially mapping it. Good review of logistic regression approach for mapping streamflow presence and absence. - Introduction would benefit by discussion of how streamflow intermittence varies by stream order in arid vs humid locations - Authors asssert that climate variables constant throughout catchment, so the authors can focus on geological and pedologial factors, but is this proven? pointed to another paper Pfister et al. (2017) for major assertion that climate does not significantly vary across the catchment, better to include figure in paper showing this. - Discussion could benefit from greater synthesis on process-based logic behind the significance of certain variables being included in wet vs dry vs annual models - I appreciate that streamflow intermittence is not easy to predict. Perhaps including gridded estimates of precipitation at observed timesteps would help improve performance. I expect that this data is available. Could use precipitation on day of flow observation as well as 1 day prior , or 7-day antecedent precipitation for example. The current models of landscape / soil / geology variables are justified, but including climate could potentially improve performance substantially.

Technical comments:

- Instead of "permanent" term, suggest "perennial" throughout. Also use this instead of "continuous" - Provide additional information on the range in climatic conditions when virtual sites were visited and how exact locations were chosen - relative intermittency of streamflow Ir is analogous to more commonly used "no flow fraction" - Clarification of local and catchment variables needed in abstract - Better clarification in abstract of which variables important in which models (annual, wet, dry) - Final sentence of abstract: could suggest that the first step was the extensive monitoring that was

completed using a variety of sensors - Combine first two sentences of introduction - Figure 5, color bar legend does not match figure caption - First sentence of conclusion, highlight the novelty of this approach as compared to previous logistic regression approaches mentioned in intro - Conclusion a bit redundant with discussion, suggest focusing on key takeaways

—————————————————

---

## Referee Comment (RC2) · Anonymous Referee #2 · 13 Jul 2020

This study presets a method to estimate streamflow intermittency across the temperate landscape. The method builds upon previous work adding a probability estimation. The end goal is to get towards spatially explicit mapping of streams to support representation in modeling efforts. The study focuses on the well-investigated Attert Catchment in Luxembourg. The topic is timely given potential extremes and change brought about be climate shifts in landscape-scale hydrological function. The paper is well written and easy to follow. With that, there are only a few comments that need to be addressed in a revision. Addressing these will take some work, but nothing too laborious and should strengthen the study.

The first aspect that should be address would be the relative coarse resolution of the DEM of 15 m resolution impact on model uncertainty and/or sensitivity. The authors

point out the potential shortcoming due to the resolution and highlight the no higher resolution is available. It still seems more is needed here to support the potential role of the coarse resolution. For example, missing even the smallest of surface channel flow to connect sections of flow streams is problematic for the modeling approach. So, of course, I'm left wondering on the impact for the modeling and estimation. Could there be some quantification of the potential uncertainty? For example, a pseudo/synthetic reduction (or increase) in coarseness of the resolution and a simple re-run to assess the change in accuracy? That could at least partly quantify impacts and try to put a number on it. I'm sure there are plenty of more creative alternatives, but putting some uncertainty bound (confidence) around your estimates as a function of the spatial resolution would be helpful since you are thinking to use your model output in connection with a modeling effort. Without tracking through the added uncertainty, you're likely to see some huge multiplier effects for follow up estimations.

The other aspect the jumps at me would be the lack of some validation using a leave on out or a split sampling to get at model sensitivity and robustness. There is much done to assess the model performance and consider the power of each separate model. But, could you provide some sort of validation of the accuracy? Seems a systematic leave-one-out-at-a-time approach could be useful to model a real data point and see if you got it right of wrong (and track the false positives to see if you are getting wet or dry too much). Alterative could be some Monte Carlo split samples to estimate several points left out of a training dataset - then randomize and repeat. Yes, these approaches are brute force and cumbersome, but this study is all desktop and computer based. So should be "simple" enough to add some loops and let the program churn out some validation statistics. That would help the reader assess how much the configuration of sample locations drives the accuracy and performance. Could be you even assess the "value" of each observation point in the overall system to help future studies design where to sample intermittency for the most bang for the buck.

181, 2020.

---

## Author Comment (AC1) · 16 Aug 2020

Kaplan et al. Anonymous Referee #1 General comments: In "Predicting probabilities of streamflow intermittency across a temperate mesoscale catchment," Kaplan et al. explain the local and accumulated catchment controls on flow intermittence along the flow network of the Attert catchment in Luxembourg. Using logistic regression models for annual as well as wet and dry periods, the authors evaluate the variable importance of land cover, road density, soil, geology, and terrain metrics in controlling flow intermittence. The authors use a unique, empirical high spatial and temporal dataset (Kaplan et al., 2019) to develop these models. The authors presentation of model results and discussion of implications and uncertainties is fairly robust but with several opportunities for enhancement prior to publication. In addition, there are grammatical and tense errors and inconsistency throughout, so the manuscript needs detailed proofreading prior to publication. This study is an important scientific contribution with high scientific and presentation quality Dear Referee, Thank you for your helpful comments and questions to our manuscript. We are happy to see your positive summary of our work. Please find your questions and comments marked as e.g. « R1.C1: question/comment» followed by our answer marked as e.g. R1.A1: below. Best regards, Kaplan et al.

«R1.C1. General comments: [. . .] there are grammatical and tense errors and inconsistency throughout [. . .] »

R1.A1 Thank you for pointing this out. We will make sure to do another thorough correction of the manuscript.

«R1.C2. Specific comments: Introduction does a good job of describing prior work and science gaps in the field of hydrology relevant to studying drivers of intermittent flow occurrence as well as spatially mapping it. Good review of logistic regression approach for mapping streamflow presence and absence. - Introduction would benefit by discussion of how streamflow intermittence varies by stream order in arid vs humid locations »

R1.A2.

We will add the following sentence to the introduction at page 1, line 26: These streams are largely controlled by the climatic conditions with generally low but spatially highly variable precipitation as well as high rates of direct evaporation and evapotranspiration through plants (Datry et al., 2017). However, in temperate regions the occurrence of intermittent streams is commonly limited to headwaters, and the wetter climate generally provides enough overland flow and groundwater recharge to maintain perennial rivers for large parts of the river system (Jaeger et al., 2017). These intermittent streams in

temperate regions have only recently gotten more attention [. . .].

References update: Datry T., Bondana, N. and Boulton A.J.: Chapter 1 - General introduction. In: Datry T., Bondana, N. and Boulton A.J.: Intermittent Rivers and Ephemeral Streams – Ecology and Management. Academic Press, doi: https://doi.org/10.1016/B978-0-12-803835-2.00001-2, 2017.

Jaeger K.L., Sutfin N.A., Tooth S., Mechaelides K. and Singer M.: Chapter 2.1 – Geomorphology and Sediment Regimes of Intermittent Rivers and Ephemeral Streams. In: Datry T., Bondana, N. and Boulton A.J.: Intermittent Rivers and Ephemeral Streams – Ecology and Management. Academic Press, doi: https://doi.org/10.1016/B978-0-12-803835-2.00002-4, 2017.

«R1.C3. Specific comments: Authors asssert that climate variables constant throughout catchment, so the authors can focus on geological and pedologial factors, but is this proven? pointed to another paper Pfister et al. (2017) for major assertion that climate does not significantly vary across the catchment, better to include figure in paper showing this. »

R1.A3.

To show that rainfall variability is not a major control in this catchment we will add two figures to the supplement. On the one hand the rainfall maps, and on the other hand local rainfall plotted against the local residuals of the statistical models (at the measurement points).

Full caption Fig. S2:

Figure S2: Cumulative Precipitation distribution in the Attert catchment for the annual period July 2016 to July 2017 (a), the wet period (February to April, (b)) and dry period (June to August, (c)). Note: wet and dry here refers to discharge, not to rainfall input. Precipitation data is interpolated with ordinary kriging from site specific local precipitation data (black stars). Precipitation data was provided from a precipitation modelling

approach by Neuper & Ehret (2019) which combines weather radar and ground-based precipitation data. The deviation between observed and modelled intermittency is plotted for the corresponding periods.

Figure S2 shows a trend of the precipitation decreasing from North/North-West to South/South-East. This trend shows a difference in annual precipitation of about 100 mm which is around 25% of the maximum precipitation sum. The wet period turned out to show lower precipitation sums than the dry period, possibly due to a rainfall data gap of one week and the wet period being 3 days shorter than the dry period. The absolute difference between the minimum and maximum precipitation in the wet period is only 17mm (25% of the maximum). During the dry period the difference in precipitation between driest (130mm) and wettest (195mm) regions amounts to 33% of the maximum. This trend is roughly reflected by the geological setting. The North/North-Western part of the catchment comprises the highest ridges of the catchment with slate geology. The precipitation in this part of the catchment is higher through the orographic uplift at the ridge, whereas the lower areas of the catchment (mainly the geologies of marls and sandstone) lie in the rain shadow of the Ardennes (Neuper & Ehret, 2019). However, the precipitation sums do not reflect the runoff behavior in the corresponding periods. The wet period shows lower precipitation sums compared to the dry period. This indicates that evapotranspiration is a stronger control of runoff in the catchment, which is also stated by Wrede et al. (2014). On the other hand, ephemeral streams which are controlled by intense precipitation events are not necessarily dependent on the precipitation sums during a specific period, but stronger linked to the precipitation intensity during a single event (Datry et al., 2017). The residuals between observed and modelled intermittency cannot be explained by the precipitation patterns (Fig. S2 and S3), which confirms our assumption that rainfall patterns do not need to be included as a predictor in the context of this study. However, we will investigate the dynamics of patterns of intermittency and their dynamic controls in a separate study.

We will provide a reference to the supplementary Figure S2 in the main manuscript on
page 4 line 20: "[. . .] results in similar climate conditions across the catchment (Pfister et al., 2017, Fig. S2)."

We will also provide a reference to the supplementary Figure S3 in the main manuscript on page 22 line 13: [. . .] which is supported by the small differences in annual precipitation (Pfister et al., 2005; Wrede et al., 2014, Fig. S3).

Reference Update in Supplements:

Neuper M. and Ehret U.: Quantitative precipitation estimation with weather radar using a data- and information-based approach. Hydrol. Earth Syst. Sci., 23, 3711–3733, https://doi.org/10.5194/hess-23-3711-2019, 2019.

«R1.C4. Specific comments: I appreciate that streamflow intermittence is not easy to predict. Perhaps including gridded estimates of precipitation at observed timesteps would help improve performance. I expect that this data is available. Could use precipitation on day of flow observation as well as 1 day prior , or 7-day antecedent precipitation for example. The current models of landscape / soil / geology variables are justified, but including climate could potentially improve performance substantially. »

R1.A4.

Using event-based precipitation and the pre-event precipitation as indicator for the wetness of the system is indeed a good suggestion. However, the study at hand focuses on a full year and two three month periods and does not drill down to the event-scale. As the precipitation patterns are unable to explain the spatial pattern of the model residuals we think that at it is a viable assumption to neglect the rainfall variability at this 3-month or annual time scale and to focus on quasi-static landscape controls. The dynamics of intermittency, also at the event scale are being tackled in a separate study.

«R1.C5. Technical comments: Instead of "permanent" term, suggest "perennial" throughout. Also use this instead of "continuous »

R1.A5.

We will use the term perennial as you suggest. This will be possible due to our definition of perennial also for the 3 month period at page 24, line 11. We will keep the term "permanent" at......page 26, line 27 because we use it here as the description of the probability range of streamflow. ...page 2, line 11 because the perennial classification is defined for rivers and not for the baseflow component.

We will keep the term "continuous" for page 2, line 13 as this describes a time period not the annual behavior. We will change the term from "continuous" to "perennial" at page 6 line 26

«R1.C6. Technical comments: Provide additional information on the range in climatic conditions when virtual sites were visited and how exact locations were chosen »

R1.A6.

We will add this section on page 6, line 17: The majority of virtual sites were visited every two months during maintenance campaigns for the monitoring sites. Virtual sites located at the ridge of southern sandstone regions were visited less frequent, but showed no sign of surface flow during all visits. The sites were added to the dataset at locations which were a) frequently visited and thus known to have no-flow behavior and b) also in areas where no-flow observations are underrepresented in the dataset, such as ridges in the sandstone region or the riparian zone of valleys in the slate region and c) the capability of the model would be enhanced.

«R1.C7. Technical comments: relative intermittency of streamflow Ir is analogous to more commonly used "no flow fraction" »

R1.A7.

Unfortunately, our literature search for the term "no flow fraction" for what we called "relative intermittency" was unsuccessful. We would therefore leave the decision if this is the more common term and should be used to the editor: if the editor confirms this, we are happy to change the terminology. However, in that case we would find it helpful

to be pointed to the publications using this term.

«R1.C8. Technical comments: Clarification of local and catchment variables needed in abstract »

R1.A8.

We will add the following sentence on page 1 line 16: "We used 15 spatial predictors describing land cover, track (road) density, terrain metrics, soil and geological properties. Predictors were included as local scale information, represented by the local value at the catchment outlet and as integral catchment information calculated as the mean catchment value over all pixels upslope of the catchment outlet."

«R1.C9. Technical comments: Better clarification in abstract of which variables important in which models (annual, wet, dry) »

R1.A9.

We will change the section page 1, line 16 – 18 from:

"The terrain metrics catchment area and profile curvature were the most important predictors for all models. However, the models which include the dry period of the year reveal the importance of soil hydraulic conductivity, bedrock permeability and in case of the annual model the presence of tracks (roads) during low flow conditions."

To

"The terrain metrics catchment area and profile curvature were identified in all models as the most important predictors, and the model for the wet period was based solely on these two predictors. However, the model for the dry period additionally comprises soil hydraulic conductivity and bedrock permeability. The annual model with the most complex predictor set contains the predictors of the dry period model plus the presence of tracks (roads)."

«R1.C10. Technical comments: Final sentence of abstract: could suggest that the first

step was the extensive monitoring that was completed using a variety of sensors »

R1.A10.

We will change the sentence to: This approach, based on extensive monitoring and statistical modeling, is a first step to provide detailed spatial information for hydrological modelling as well as management practice.

«R1.C11. Technical comments: Combine first two sentences of introduction »

R1.A11.

We will change the sentence to: "Even though intermittent streams and rivers represent more than half of the global stream network (Datry et al., 2014) they have been studied to a far lesser degree than their perennial counterparts."

«R1.C12. Technical comments: Figure 5, color bar legend does not match figure caption »

R1.A12.

The color bar refers to the plotted points (color coded). The first sentence in the caption refers to the color bar. The second sentence in the caption refers to the gray boxes in the plot. We will change this sentence to: "The grey boxes indicate the classes of ephemeral (Ir < 0.1), intermittent ($0.1 \leq$ Ir < 0.8) and perennial ($0.8 \leq$ Ir < 1.0) streamflow."

«R1.C13. Technical comments: First sentence of conclusion, highlight the novelty of this approach as compared to previous logistic regression approaches mentioned in intro »

R1.A13.

We will change the sentence from:

"This study presents a novel approach of modelling streamflow intermittency using

logistic regression models with data of streamflow presence/absence and spatial predictors."

to:

"This study presents a novel approach of modelling streamflow intermittency using logistic regression models. In contrast to earlier studies we use the newly introduced response variable of relative intermittency instead of binary streamflow classes (e.g. intermittent/perennial) which allows for modelling of streamflow probabilities. The comparable climatic conditions across the studied catchment permit a focus on quasi-static predictor variables such as geology, soil, terrain, land cover or tracks and roads."

«R1.C14. Technical comments: Conclusion a bit redundant with discussion, suggest focusing on key takeaways »

R1.A14.

We will revise the conclusions thoroughly to improve the focus.
* * *
(a)

(b)

(c)

0   3.75   7.5   15
Kilometers

>50          <67

0   3.75   7.5   15
Kilometers

0   3   6   12
Kilometers

\* Local Precipitation Data

Deviation Observed vs. Modelled Intermittency

Precipitation

>328        360        390        <438

<-0.75   -0.75 - -0.50   -0.50 - -0.25   -0.25 - -0.0   0.0 - 0.25   0.25 - 0.50   0.50 - 0.75   >0.75

>130          <195

0   3.75   7.5   15
Kilometers

**Fig. 1.** Figure S2: Cumulative Precipitation distribution in the Attert catchment for the annual period July 2016 to July 2017 (a), the wet period (February to April, (b)) and dry period (June to August, (c)).

[Figure]

**Fig. 2.** Figure S3: Deviation between observed and modelled plotted against the corresponding precipitation sums of the modeled periods.

---

## Author Comment (AC2) · 16 Aug 2020

Kaplan et al. Anonymous Referee #2 Received and published: 13 July 2020

«This study presets a method to estimate streamflow intermittency across the temperate landscape. The method builds upon previous work adding a probability estimation. The end goal is to get towards spatially explicit mapping of streams to support representation in modeling efforts. The study focuses on the well-investigated Attert Catchment in Luxembourg. The topic is timely given potential extremes and change brought about be climate shifts in landscape-scale hydrological function. The paper is

well written and easy to follow. With that, there are only a few comments that need to be addressed in a revision. Addressing these will take some work, but nothing too laborious and should strengthen the study.»

Dear Referee,

Thank you for your helpful comments and suggestions to improve our manuscript. We appreciate your general agreement with the structure of the paper and content of our manuscript. Please find your questions and comments marked as e.g. « R2.C1: question/comment» followed by our answer marked as e.g. R2.A1: below. Best regards, Kaplan et al.

«R2.C1. Specific comments: The first aspect that should be address would be the relative coarse resolution of the DEM of 15 m resolution impact on model uncertainty and/or sensitivity. The authors point out the potential shortcoming due to the resolution and highlight the no higher resolution is available. It still seems more is needed here to support the potential role of the coarse resolution. For example, missing even the smallest of surface channel flow to connect sections of flow streams is problematic for the modeling approach. So, of course, I'm left wondering on the impact for the modeling and estimation. Could there be some quantification of the potential uncertainty? For example, a pseudo/synthetic reduction (or increase) in coarseness of the resolution and a simple re-run to assess the change in accuracy? That could at least partly quantify impacts and try to put a number on it. I'm sure there are plenty of more creative alternatives, but putting some uncertainty bound (confidence) around your estimates as a function of the spatial resolution would be helpful since you are thinking to use your model output in connection with a modeling effort. Without tracking through the added uncertainty, you're likely to see some huge multiplier effects for follow up estimations »

R2.A1.

Good point. It was indeed unfortunate that no higher resolution DEM was available! However, the DEM used in this study is suitable for most of the sites also after careful Interactive comment
shifting of some sites to the calculated stream channel. Nevertheless, 8 sites may be prone to non-accurate delineation of the catchment area, mainly in flat and/or areas with highly detailed relief and small drainage channels. We will make an effort to better assess the uncertainty caused by the coarse-resolution DEM and will include this in the revised manuscript.

«R1.C2. Specific comments: The other aspect the jumps at me would be the lack of some validation using a leave on out or a split sampling to get at model sensitivity and robustness. There is much done to assess the model performance and consider the power of each separate model. But, could you provide some sort of validation of the accuracy? Seems a systematic leave-one-out-at-a-time approach could be useful to model a real data point and see if you got it right of wrong (and track the false positives to see if you are getting wet or dry too much). Alterative could be some Monte Carlo split samples to estimate several points left out of a training dataset - then randomize and repeat. Yes, these approaches are brute force and cumbersome, but this study is all desktop and computer based. So should be "simple" enough to add some loops and let the program churn out some validation statistics. That would help the reader assess how much the configuration of sample locations drives the accuracy and performance. Could be you even assess the "value" of each observation point in the overall system to help future studies design where to sample intermittency for the most bang for the buck. »

**R2.A2.**

We will add the Leave One Out Cross Validation (LOOCV) approach and update the Methods section for the map comparison. Therefor we will change the following parts in the manuscript:

Methods: page 15 line 4:

[...] Due to the small data set which does not allow for a split validation approach, a leave on out cross validation (LOOCV) approach (e.g. Akbar et al., 2019; Ossa-Moreno
et al., 2019) was applied to validate the model based on the original data set. Therefore, for each observation n in the dataset one model is run with one observation left out from the dataset. Thereafter, the GLM derived from n-1 data points is used to predict the value y\_hat for the left-out point y. This process is repeated for all observations. The measure of Root Mean

Square Error RMSE is used to assess the model accuracy as follows:

RMSE=  $\sqrt{(1/n \sum_{i=1}^{n} (y_i - (y_i hat)))}$

and bias of the model by:

 $Bias=1/n \sum_{i=1}^{n} (y_i - (y_i hat))$

where n is the number of observations, y\_hat is the predicted and y the observed relative intermittency (Akbar et al., 2019).

Methods: page 15 line 8:

In order to validate the results from the classified reaches we compare the stream length of the modelled streams with the length of the streams from the topographic map (Le Gouverment du Grand-Duché de Luxembourg, 2009). We assume that the mapped stream network approximately represents the natural layout of the stream network in areas with lower human impacts.

On page 17, line 4 (Results section):

The RMSE for the Model-Y is with 0.26 the lowest among all models with RMSE of 0.263 and 0.29 for the Models W2 and D2 respectively. The bias of the models is very low and ranges around zero with values between - 9.6 \* 10-4 (Model Y) and 4 \* 10-5 (Model W2). Model deviations for the models Y, D2 and W2 is shown in fig. 7.

On page 23 line 33 (discussion section): Instead of:

"As the distribution of measurement sites in the data set has a strong tendency towards
permanent streamflow sites and thus to the perennial reaches, this leads to an underrepresentation of the intermittent and ephemeral reaches in the data when splitting the data for model evaluation (Figure 6). We therefore evaluated the model by its ability to predict the spatial distribution of intermittent/perennial streams compared to the mapped stream network. We assume that the mapped stream network approximately represents the natural layout of the stream network in areas with lower human impacts. However, alteration of the natural stream network in areas of artificial and agricultural land use can be severe and thus misleading when comparing to model results."

we will write:

"[...] good GLM setup. However, the leave- one-out cross validation allows for a databased validation. The RMSE values (0.26 - 0.31) obtained for the different models related to the maximum possible RMSE of 1 shows overall model deviations of around 26 to 30 %. The plotted deviations (fig. 7) reveal some extreme deviations of nearly 1. The majority of perennial streams seem to be well represented by the model, while many of the ephemeral streams have deviations of > 0.5. This could be due to the distribution of observation sites in the data set, which have a strong tendency towards permanent streamflow sites and thus to the perennial reaches, while intermittent and ephemeral reaches are underrepresented (Figure 6). The better representation of perennial streams becomes also visible in the model validation by its ability to predict the spatial distribution of intermittent/perennial streams compared to the mapped stream network."

Fig. 7 Full caption:

Figure 7: Model deviations for all models. The indicated intermittency class is based on the classification scheme from Hedman & Osterkamp (1982). The observed relative intermittency is

Akbar S., Kathuria A. and Maheshwari B.: Combining imaging techniques with nonparametric modelling to predict seepage hotspots in irrigation channels. Irrigation Science, 37, 11–23, doi: 10.1007/s00271-018-0596-6, 2019.

Ossa-Moreno J., Keir G., McIntyre N., Cameletti M. and Rivera D.: Comparison of approaches to interpolating climate observations in steep terrain with low-density gauging networks, Hydrol. Earth Syst. Sci., 23, 4763–4781 doi: 10.5194/hess-23-4763-2019, 2019.

**HESSD**
**HESSD**
**Fig. 1.** Figure 7: Model deviations for all models. The indicated intermittency class is based on the classification scheme from Hedman & Osterkamp (1982). The observed relative intermittency is

---

## Author Response (AR1)

General comments:

In "Predicting probabilities of streamflow intermittency across a temperate mesoscale catchment," Kaplan et al. explain the local and accumulated catchment controls on flow intermittence along the flow network of the Attert catchment in Luxembourg. Using logistic regression models for annual as well as wet and dry periods, the authors evaluate the variable importance of land cover, road density, soil, geology, and terrain metrics in controlling flow intermittence. The authors use a unique, empirical high spatial and temporal dataset (Kaplan et al., 2019) to develop these models. The authors presentation of model results and discussion of implications and uncertainties is fairly robust but with several opportunities for enhancement prior to publication. In addition, there are grammatical and tense errors and inconsistency throughout, so the manuscript needs detailed proofreading prior to publication. This study is an important scientific contribution with high scientific and presentation quality

Dear Referee,

Thank you for your helpful comments and questions to our manuscript. We are happy to see your positive summary of our work. Please find your questions and comments marked as e.g. << R1.C1: question/comment>> followed by our answer marked as e.g. R1.A1: below.

Best regards,

Kaplan et al.

<<R1.C1.
General comments:
[…] there are grammatical and tense errors and inconsistency throughout […]

>>

R1.A1

We made another thorough correction of the manuscript.

<<R1.C2.
Specific comments:

Introduction does a good job of describing prior work and science gaps in the field of hydrology relevant to studying drivers of intermittent flow occurrence as well as spatially mapping it. Good review of logistic regression approach for mapping streamflow presence and absence. - Introduction would benefit by discussion of how streamflow intermittence varies by stream order in arid vs humid locations
>>

R1.A2.

We added the following sentence to the introduction at page 1, line 26:

These streams are largely controlled by the climatic conditions with generally low but spatially highly variable precipitation as well as high rates of direct evaporation and evapotranspiration through plants (Datry et al., 2017). However, in temperate regions the occurrence of intermittent streams is commonly limited to headwaters, and the wetter climate generally provides enough overland flow and groundwater recharge to maintain perennial rivers for large parts of the river system (Jaeger et al., 2017). These intermittent streams in temperate regions have only recently gotten more attention […].

We updated the references:

Datry T., Bondana, N. and Boulton A.J.: Chapter 1 - General introduction. In: Datry T., Bondana, N. and Boulton A.J.: Intermittent Rivers and Ephemeral Streams – Ecology and Management. Academic Press, doi: https://doi.org/10.1016/B978-0-12-803835-2.00001-2, 2017.

Jaeger K.L., Sutfin N.A., Tooth S., Mechaelides K. and Singer M.: Chapter 2.1 – Geomorphology and Sediment Regimes of Intermittent Rivers and Ephemeral Streams. In: Datry T., Bondana, N. and Boulton A.J.: Intermittent Rivers and Ephemeral Streams – Ecology and Management. Academic Press, doi: https://doi.org/10.1016/B978-0-12-803835-2.00002-4, 2017.

<<R1.C3.
Specific comments:
Authors asssert that climate variables constant throughout catchment, so the authors can focus on geological and pedologial factors, but is this proven? pointed to another paper Pfister et al. (2017) for major assertion that climate does not significantly vary across the catchment, better to include figure in paper showing this. >>

R1.A3.

To show that rainfall variability is not a major control in this catchment we added two figures to the supplement. On the one hand the rainfall maps, and on the other hand local rainfall plotted against the local residuals of the statistical models (at the measurement points).

We added the following section to the supplement:

**"S2 Precipitation distribution in the Attert catchment**

Spatial variability of precipitation is not a major control in the Attert catchment. Figure S2 shows the spatial distribution of precipitation for the modelled time periods. Figure S3 shows the local precipitation at the measurement points plotted against the local residuals of the statistical models."

[Figure]

**Figure S2: Cumulative Precipitation distribution in the Attert catchment for the annual period July 2016 to July 2017 (a), the wet period (February to April, (b)) and dry period (June to August, (c)). Note: wet and dry here refers to discharge, not to rainfall input. Precipitation data is interpolated with ordinary kriging from site specific local precipitation data (black stars). Precipitation data was provided from a precipitation modelling approach by Neuper & Ehret (2019) which combines weather radar and ground-based precipitation data. The deviation between observed and modelled intermittency is plotted for the corresponding periods.**

[Figure]

**Figure S3: Deviation between observed and modelled plotted against the corresponding precipitation sums of the modeled periods.**

We provided a reference to the supplementary Figure S2 in the main manuscript on page 4 line 20:

"[…] results in similar climate conditions across the catchment (Pfister et al., 2017; Fig. S2)."

We provided a reference to the supplementary Figure S3 in the main manuscript on page 22 line 13:

[…] which is supported by the small differences in annual precipitation (Pfister et al., 2005; Wrede et al., 2014; Fig. S3).

We updated the references in the supplements:

Neuper M. and Ehret U.: Quantitative precipitation estimation with weather radar using a data- and information-based approach. Hydrol. Earth Syst. Sci., 23, 3711–3733, https://doi.org/10.5194/hess-23-3711-2019, 2019.

<<R1.C4.
Specific comments:
I appreciate that streamflow intermittence is not easy to predict. Perhaps including gridded estimates of precipitation at observed timesteps would help improve performance. I expect that this data is available. Could use precipitation on day of flow observation as well as 1 day prior, or 7-day antecedent precipitation for example. The current models of landscape / soil / geology variables are justified, but including climate could potentially improve performance substantially.
>>

R1.A4.

According to our previous statement in the review reply:

Using event-based precipitation and the pre-event precipitation as indicator for the wetness of the system is indeed a good suggestion. However, the study at hand focuses on a full year and two three-month periods and does not drill down to the event-scale. As the precipitation patterns are unable to explain the spatial pattern of the model residuals we think that at it is a viable assumption to neglect the rainfall variability at this 3-month or annual time scale and to focus on quasi-static landscape controls. The dynamics of intermittency, also at the event scale are being tackled in a separate study.

We leave it to the additional information provided for R1.C3.

<<R1.C5.
Technical comments:
Instead of "permanent" term, suggest "perennial" throughout. Also use this instead of "continuous
>>

R1.A5.

We used the term perennial as you suggest.

We kept the term "permanent" on…

…page 26, line 27 because we use it here as the description of the probability range of streamflow.

…page 2, line 11 because the perennial classification is defined for rivers and not for the baseflow component.

We changed the term "permanent" to "perennial" on…

…page 7, line 6

…page 18, line 21

… page 28, line 6

…page 30, line 7

We kept the term "continuous" for page 2, line 13 as this describes a time period not the annual behavior.

We changed the term from "continuous" to "perennial" on page 6 line 26

<<R1.C6.
Technical comments:
Provide additional information on the range in climatic conditions when virtual sites were visited and how exact locations were chosen
>>

R1.A6.

We added this section on page 6, line 17:

The majority of virtual sites were visited every two months during maintenance campaigns for the monitoring sites. Virtual sites located at the ridge of southern sandstone regions were visited less frequently, but showed no sign of surface flow during all visits. The sites were added to the dataset at locations which a) were frequently visited and thus known to have no-flow behavior and b) also in areas where no-flow observations are underrepresented in the dataset, such as ridges in the sandstone region or the riparian zone of valleys in the slate region and c) improved the model.

<<R1.C7.
Technical comments:
relative intermittency of streamflow Ir is analogous to more commonly used "no flow fraction"
>>

R1.A7.

Referring to our previous reply to the referee comment:

"Unfortunately, our literature search for the term "no flow fraction" for what we called "relative intermittency" was unsuccessful. We would therefore leave the decision if this is the more common term and should be used to the editor: if the editor confirms this, we are happy to change the terminology. However, in that case we would find it helpful to be pointed to the publications using this term."

In absence of further recommendations to apply changes to this terminology we leave it as is.

<<R1.C8.
Technical comments:
Clarification of local and catchment variables needed in abstract
>>

R1.A8.

We added the following sentence on page 1 line 16:

"We used 15 spatial predictors describing land cover, track (road) density, terrain metrics, soil and geological properties. Predictors were included as local scale information, represented by the local value at the catchment outlet and as integral catchment information calculated as the mean catchment value over all pixels upslope of the catchment outlet.

<<R1.C9.
Technical comments:
Better clarification in abstract of which variables important in which models (annual, wet, dry)
>>

R1.A9.

We changed the section page 1, line 16 – 18 from:

"The terrain metrics catchment area and profile curvature were the most important predictors for all models. However, the models which include the dry period of the year reveal the importance of soil hydraulic conductivity, bedrock permeability and in case of the annual model the presence of tracks (roads) during low flow conditions."

To

"The terrain metrics catchment area and profile curvature were identified in all models as the most important predictors, and the model for the wet period was based solely on these two predictors. However, the model for the dry period additionally comprises soil hydraulic conductivity and bedrock permeability. The annual model with the most complex predictor set contains the predictors of the dry period model plus the presence of tracks."

<<R1.C10.
Technical comments:
Final sentence of abstract: could suggest that the first step was the extensive monitoring that was completed using a variety of sensors
>>

R1.A10.

We changed the sentence to:

This approach, based on extensive monitoring and statistical modeling, is a first step to provide detailed spatial information for hydrological modelling as well as management practice.

<<R1.C11.
Technical comments:

Combine first two sentences of introduction
>>

R1.A11.

We changed the sentence to:

"Even though intermittent streams and rivers represent more than half of the global stream network (Datry et al., 2014) they have been studied to a far lesser degree than their perennial counterparts."

<<R1.C12.
Technical comments:
Figure 5, color bar legend does not match figure caption
>>

R1.A12.

The color bar refers to the plotted points (color coded). The first sentence in the caption refers to the color bar. The second sentence in the caption refers to the gray boxes in the plot. We changed this sentence to:

"The grey boxes indicate the classes of ephemeral ($Ir < 0.1$), intermittent ($0.1 \leq Ir < 0.8$) and perennial ($0.8 \leq Ir < 1.0$) streamflow."

<<R1.C13.
Technical comments:
First sentence of conclusion, highlight the novelty of this approach as compared to previous logistic regression approaches mentioned in intro
>>

R1.A13.

We changed the sentence from:

"This study presents a novel approach of modelling streamflow intermittency using logistic regression models with data of streamflow presence/absence and spatial predictors."

to:

"This study presents a novel approach of modelling streamflow intermittency using logistic regression models. In contrast to earlier studies we use the here newly introduced response variable of relative intermittency instead of binary streamflow classes (e.g. intermittent/perennial) which allows for modelling of streamflow probabilities. The comparable climatic conditions across the studied catchment permit a focus on quasi-static predictor variables such as geology, soil, terrain, land cover or tracks and roads."

<<R1.C14.
Technical comments:
Conclusion a bit redundant with discussion, suggest focusing on key takeaways
>>

R1.A14.

We changed the following sentences and sections:

Models for the wet periods were mainly driven by the terrain metrics contributing area and profile curvature. (page 30, line 16)

to

Models for the wet periods were mainly driven by the terrain metrics contributing area and profile curvature which represent a measure for saturation probability.

Additional predictors of saturated conductivity and relative bedrock permeability that define the transmissivity and the storage capacity of the system as well as the track density as potential indicator of local Hortonian overland flow reveal a system of higher hydrological complexity during periods of low streamflow. Furthermore, the selection of predictors shows the viability of the innovative approach using integrated contributing area information for certain predictors such as effective saturated conductivity and relative bedrock permeability. Both predictors contribute significantly to the models for the dry periods. This indicates that the loss of water through high infiltration capacity and storage capacity in the upstream contributing area are among the controlling factors of streamflow intermittency. (page 30, line 24)

to

Dry period models contained relative bedrock permeability and conductivity as additional predictors, which are a measure of transmissivity and storage capacity of the system in the dry system state. The model for the annual period includes all the predictors from the dry period and additionally track density, which was recognized as potential indicator of local Hortonian overland flow. The innovative approach using integrated contributing area information for the predictors of saturated hydraulic conductivity and bedrock permeability was valuable to describe upstream controls of intermittency like infiltration and storage capacity.

Modelling results are classified into ephemeral for streamflow presence of less than 10%, intermittent between 10% and 80% and perennial for more than 80% within a defined time period. Based on this classification all models are able to discriminate between intermittent and perennial streams.(page 31, line 4)

to

After classification into ephemeral, intermittent and perennial reaches all models are able to discriminate between intermittent and perennial streams.

We deleted the sentence:

The high correlation of catchment area on log scale to the topographic wetness index (TWI) indicates that the probability of saturation is an important driver in wet periods. (page 30, line 18)

*R 2*
This study presets a method to estimate streamflow intermittency across the temperate landscape. The method builds upon previous work adding a probability estimation. The end goal is to get towards spatially explicit mapping of streams to support representation in modeling efforts. The study focuses on the well-investigated Attert Catchment in Luxembourg. The topic is timely given potential extremes and change brought about be climate shifts in landscape-scale hydrological function. The paper is well written and easy to follow. With that, there are only a few comments that need to be addressed in a revision. Addressing these will take some work, but nothing too laborious and should strengthen the study.

Dear Referee,

Thank you for your helpful comments and suggestions to improve our manuscript. We appreciate your general agreement with the structure of the paper and content of our manuscript. Please find your questions and comments marked as e.g. << R1.C1: question/comment>> followed by our answer marked as e.g. R1.A1: below.

Best regards,

Kaplan et al.

<<R1.C1.
Specific comments:
The first aspect that should be address would be the relative coarse resolution of the DEM of 15 m resolution impact on model uncertainty and/or sensitivity. The authors point out the potential shortcoming due to the resolution and highlight the no higher resolution is available. It still seems more is needed here to support the potential role of the coarse resolution. For example, missing even the smallest of surface channel flow to connect sections of flow streams is problematic for the modeling approach. So, of course, I'm left wondering on the impact for the modeling and estimation. Could there be some quantification of the potential uncertainty? For example, a pseudo/synthetic reduction (or increase) in coarseness of the resolution and a simple re-run to assess the change in accuracy? That could at least partly quantify impacts and try to put a number on it. I'm sure there are plenty of more creative alternatives, but putting some uncertainty bound (confidence) around your estimates as a function of the spatial resolution would be helpful since you are thinking to use your model output in connection with a modeling effort. Without tracking through the added uncertainty, you're likely to see some huge multiplier effects for follow up estimations
>>

We marked 9 sites with increased uncertainty in terms of catchment delineation in Figure 1. We updated the following sentence to the figure caption:

"Sites marked with a purple triangle have an increased uncertainty concerning the delineation of the catchment area."

In the methods section (page 9, line 11) we added the sentence:

For a few measurement locations, the relatively coarse DEM resulted in uncertainty in the delineation of catchment area (see Figure 1).

In the discussion section (page 29, line 25) we added the sentence:

In the dataset of this study 9 sites may be prone to non-accurate delineation of the catchment area, mainly in areas with very flat or highly detailed relief (Fig. 1).

In the following sentence we change the word "However" to "Unfortunately"

<<R1.C1.
Specific comments:
The other aspect the jumps at me would be the lack of some validation using a leave on out or a split sampling to get at model sensitivity and robustness. There is much done to assess the model performance and consider the power of each separate model. But, could you provide some sort of validation of the accuracy? Seems a systematic leave-one-out-at-a-time approach could be useful to model a real data point and see if you got it right of wrong (and track the false positives to see if you are getting wet or dry too much). Alterative could be some Monte Carlo split samples to estimate several points left out of a training dataset - then randomize and repeat. Yes, these approaches are brute force and cumbersome, but this study is all desktop and computer based. So should be "simple" enough to add some loops and let the program churn out some validation statistics. That would help the reader assess how much the configuration of sample locations drives the accuracy and performance. Could be you even assess the "value" of each observation point in the overall system to help future studies design where to sample intermittency for the most bang for the buck.

>>

We added the Leave One Out Cross Validation (LOOCV) approach and update the Methods section for the map comparison. Therefor we changed the following parts in the manuscript:

Methods: page 15 line 4:

[…] Due to the small data set which does not allow for a split validation approach, a leave on out cross validation (LOOCV) approach (e.g. Akbar et al., 2019; Ossa-Moreno et al., 2019) was chosen to validate the model based on the original data set. Thus 185 models were calibrated, each leaving out one of the data points. Then, the GLM derived from n-1 data points is used to predict the value $\hat{y}$ for the left-out point with the observed value y. This process is repeated for all observations. The measure of Root Mean Square Error RMSE is used to assess the model accuracy as follows:

$$RMSE = \sqrt{\frac{1}{n} \sum_{i=1}^{n} (y_i - \hat{y}_i)^2}$$

The bias of the model is determined by:

$$Bias = \frac{1}{n} \sum_{i=1}^{n} (y_i - \hat{y}_i)$$

[revised manuscript text omitted]

Table 1 was deleted. Thus, all following tables and their cross references were updated.

Fig. 4 and Table 4, 5, 6:

The abbreviation for relative bedrock permeability was changed from $K_r$ to $K_{br}$.

We changed an old version of the pseudo-R² to the one actually used in this study (Page 16, line 3). Thus, we could delete a second sentence mentioning the other measure (Page 16, line 24).

Page 16, line 3:

The model performance was analysed based on the Nagelkerke peudo-R² measure in order to evaluate an overall model fit but also for the ability of each model to predict intermittency classes ranging from ephemeral over intermittent to perennial.

To

The model performance was analysed based on the McFadden peudo-R² measure in order to evaluate an overall model fit but also for the ability of each model to predict intermittency classes ranging from ephemeral over intermittent to perennial.

Deleted sentence Page 16, line 24:

McFadden pseudo-R² was used as performance measure for all GLMs.

We added a description for the abbreviations used in Table 5:

"The significance of each predictor (curvature planar ($C_{pr}$), catchment area ($log(A)$), saturated hydraulic conductivity ($K_{s,avg}$), relative bedrock permeability ($K_{br}$) and track density within a 100m radius (TD100)) for each model."

We corrected the abbreviation "A" to "log(A)" in table 5 and 6.

We changed the wording from "season" to "period" on page 27, line 26 and line 31.

We corrected the words "dry" to "wet" (page 27, line 26) and "wet" to "dry" (page 27, line 34).

We updated the competing interests (page 31, line 24):

[revised manuscript text omitted]

Spatial variability of precipitation is not a major control in the Attert catchment. Figure S2 shows the spatial distribution of precipitation for the modelled time periods. Figure S3 shows the local precipitation at the measurement points plotted against the local residuals of the statistical models.

[Figure]

**Figure S2: Cumulative Precipitation distribution in the Attert catchment for the annual period July 2016 to July 2017 (a), the wet period (February to April, (b)) and dry period (June to August, (c)). Note: wet and dry here refers to discharge, not to rainfall input. Precipitation data is interpolated with ordinary kriging from site specific local precipitation data (black stars). Precipitation data was provided from a precipitation modelling approach by Neuper & Ehret (2019) which combines weather radar and ground-based**
10 **precipitation data. The deviation between observed and modelled intermittency is plotted for the corresponding periods.**

[Figure]

**Figure S3: Deviation between observed and modelled plotted against the corresponding precipitation sums of the modeled periods.**

Puraye A., Schaack A. and Faltz N.: Carte des sols du Grand-Duché de Luxembourg, 1:25.000. Feuille 4 – Esch-sur-Sûre. Ministère de l'Agriculture, de la Viticulture et des Eaux et Forêts, Administration des Services Techniques de l'Agriculture, Service de Pédologie, Ettelbruck, 1980.

Puraye A., Schaack A. and Faltz N.: Carte des sols du Grand-Duché de Luxembourg, 1:25.000. Feuille 9 – Echternach. Ministère de l'Agriculture et de la Viticulture, Administration des Services Techniques de l'Agriculture, Service de Pédologie, Ettelbruck, 1988.

30    Puraye A., Schaack A. and Faltz N.: Carte des sols du Grand-Duché de Luxembourg, 1:25.000. Feuille 6 – Beaufort. Ministère de l'Agriculture et de la Viticulture, Administration des Services Techniques de l'Agriculture, Service de Pédologie, Ettelbruck, 1995.

Puraye A., Schaack A. and Faltz N.: Carte des sols du Grand-Duché de Luxembourg, 1:25.000. Feuille 13 – Remich. Ministère de l'Agriculture et de la Viticulture, Administration des Services Techniques de l'Agriculture, Service de Pédologie, Ettelbruck, 1998.

5    Vermeire, R.: Oppervlaktegeologie en bodemgesteldheid van het westelijk Gutland (Groot-Hertogdom Luxemburg), Thèse de doctorat, Rijksuniversitéit Gent, Faculteit der Wetenschappen, 1967.

Wagener J. and Schaack A.: Carte des sols du Grand-Duché de Luxembourg, 1:25.000. Feuille 10 – Luxembourg. Ministère de l'Agriculture et de la Viticulture, Administration des Services Techniques de l'Agriculture, Service de Pédologie, 10   Ettelbruck, 1971.

Wagener J. and Schaack A.: Carte des sols du Grand-Duché de Luxembourg, 1:25.000. Feuille 1 – Troisvierges. Ministère de l'Agriculture et de la Viticulture, Administration des Services Techniques de l'Agriculture, Service de Pédologie, Ettelbruck, 1972.

Wagener J., Schaack A. and Faltz N.: Carte des sols du Grand-Duché de Luxembourg, 1:25.000. Feuille 12 – Esch-sur-Alzette. Ministère de l'Agriculture et de la Viticulture, Administration des Services Techniques de l'Agriculture, Service de Pédologie, Ettelbruck, 1975.

20   Wagener J., Vermeire R, Schaack A.: Carte des sols du Grand-Duché de Luxembourg, 1:100.000. Ministère de l'Agriculture et de la Viticulture, Administration des Services Techniques de l'Agriculture, Service de Pédologie, Ettelbruck, 1969.